# RNA atlas of human bacterial pathogens uncovers stress dynamics linked to infection

Kemal Avican [1✉], Jehad Aldahdooh[2,3], Matteo Togninalli[4,5], A. K. M. Firoj Mahmud [1], Jing Tang [2,3], Karsten M. Borgwardt[4,5], Mikael Rhen[6] & Maria Fällman [1✉]

Bacterial processes necessary for adaption to stressful host environments are potential targets for new antimicrobials. Here, we report large-scale transcriptomic analyses of 32 human bacterial pathogens grown under 11 stress conditions mimicking human host environments. The potential relevance of the in vitro stress conditions and responses is supported by comparisons with available in vivo transcriptomes of clinically important pathogens. Calculation of a probability score enables comparative cross-microbial analyses of the stress responses, revealing common and unique regulatory responses to different stresses, as well as overlapping processes participating in different stress responses. We identify conserved and species-specific 'universal stress responders', that is, genes showing altered expression in multiple stress conditions. Non-coding RNAs are involved in a substantial proportion of the responses. The data are collected in a freely available, interactive online resource (PATHOgenex).

[1] Department of Molecular Biology, Laboratory for Molecular Infection Medicine Sweden (MIMS), Umeå Centre for Microbial Research (UCMR), Umeå University, Umeå, Sweden. [2] Institute for Molecular Medicine Finland (FIMM), University of Helsinki, Helsinki, Finland. [3] Research Program in Systems Oncology, Faculty of Medicine, University of Helsinki, Helsinki, Finland. [4] Department for Biosystems Science and Engineering, ETH Zürich, Basel, Switzerland. [5] Swiss Institute for Bioinformatics, Lausanne, Switzerland. [6] Department of Microbiology, Tumor and Cell Biology (MTC), Karolinska Institute, Stockholm, Sweden. ✉email: kemal.avican@umu.se; maria.fallman@umu.se

Bacterial pathogens with different genetic and physiological features share capacity to sense and respond to external changes in the host by regulating their transcriptome. The responses are often complex, and synergistic regulation of regulatory networks can be pivotal in sensing and adapting different colonization niches during different phases of infection[1–4]. For many pathogens, the first environmental change upon infection of mammalian hosts is altered temperature. Bacteria sense changes in temperature through various sensory mechanisms triggering transcriptional changes for adaptation. The heat-shock response that aid in maintaining protein and membrane homeostasis protects bacteria from sudden temperature change[5]. Low pH in the gastrointestinal tract, genital tract, dental plaque, skin, and in phagosomes represents additional stresses[6]. *Helicobacter pylori* are successful in adaptation to acidic environments capable to colonize the stomach, which mostly relies on enzymatic activities of proteins such as urease leading to the formation of ammonia neutralizing gastric acid[7]. Another agent affecting enteric bacteria is bile salts produced in the liver and secreted to the gastrointestinal tract, as well as secondary bile salts produced by the microbial flora[8]. *Salmonella enterica* serovar Typhi can resist very high bile concentrations and can persist in gall bladder[9], where it forms biofilm on gallstones[10]. The hyperosmotic nature of blood and gastrointestinal tract can be harsh for certain pathogens, in some cases also inducing expression of virulence genes, such as in *H. pylori* and *Vibrio cholerae*[11,12]. Limited nutrient, iron, and oxygen levels are other stresses that pathogens encounter in different nishes of the host and have to adapt to for survival[13]. While free sugars are available in the blood, sugar levels can be limited in other infection sites such as respiratory tracts. Here, extracellular glycan hydrolysis is a common strategy for many bacterial pathogens to acquire nutrients[14]. Amino acid starvation can trigger the stringent reponse, mediated by guanosine tetra/pentaphosphate ((p)-ppGpp), which in turn induces stress responses for adaption to nutrient limitation. Iron is insoluble in the aerobic environment and neutral pH of serum, but invading bacteria has developed numerous mechanisms to acquire iron. One acquisition mechanism is the usage of siderophores, molecules that sequester iron and import its cargo to the bacteria through specific transporters such as the TonB/ExbB/ExbD transport system[15]. Besides natural environments of different tissues, the recruitment of immune cells, such as neutrophils and macrophages contributes to environmental changes. When activated, these cells produce toxic oxidative and nitrosative substances that bacteria have to cope with. The toxic substances are sensed by regulatory proteins such as OxyR, DksA, SsrB, OhrR, MosR, SarZ, and MgtA in different pathogens[16]. The activity of the phagocytic immune cells also consumes oxygen and contributes to local hypoxia at the infection site[17]. However, despite many known adaptation mechanisms in diverse human pathogens, there are still mechanisms and synergies between mechanisms to be identified. In addition, other parts remaining to be elucidated are the assisting interconnected regulatory networks including global and specific regulators, which can be complex involving both species-specific and shared mechanisms.

Comparative genomics studies have enabled accumulated knowledge of the diversity of bacterial pathogens. However, the question, how and when different gene products are employed by diverse pathogens to cope with the same stresses encountered in the human host, remains to be answered. To complement comparative genomics studies and answer those questions, global gene expression profiling of diverse bacterial pathogens under host-related conditions is desired. There are some resources available today covering certain bacterial species that provide some information. One is the comprehensive suite of infection-relevant conditions described for *S. enterica* where the transcriptomes are cataloged in a database[18]. Another valuable resource for retrieving information of regulatory networks associated with host cell invasion is a collection of differential expression profiles of *Salmonella* mutated in genes encoding selected transcription factors[19]. Differential gene expression profiles of *H. pylori* under five different conditions including acidic stress and growth in contact with human cell lines are also avaliable[20]. Furthermore, the BACTOME database with stationary phase expression profiles of 96 *Pseudomonas aeruginosa* clinical isolates allows linkages of phenotype, genotype, and transcriptome[21]. While these resources provide important information about gene expression and regulation under different conditions in specific bacterial pathogens, there is no resource providing information of diverse bacterial pathogens exposed to similar host-related conditions.

Here we analyzed global expression profiles of 32 different bacterial pathogens under 11 infection-relevant stress conditions. Data were collected in an interactive RNA atlas, named PATHOgenex (www.pathogenex.org), freely available to the research community. Datasets were used to uncover similarities and discrepancies in different stress responses across different groups of bacteria. This was made possible by grouping genes from different bacteria according to function and homology and computing a score showing probability to be regulated in a particular environment. These scores also allowed identifation of conserved and species-specific universal stress responders (USRs), which are genes showing altered expression in multiple stress conditions. Conserved USRs contain many known antimicrobial targets and can potentially serve as a source for studies aiming for novel targets. We also show that non-coding RNAs are differentially regulated in response to stressful environments and that novel ncRNAs can be explored from the dataset.

## Results

**Cataloging stress response of cross-microbial human pathogens**. The majority of bacteria in the PATHOgenex RNA atlas represent pathogens causing worldwide health problems. Most strains are commonly used in the microbial research community and are diverse in terms of Gram staining, phylogeny, and oxygen requirement (Fig. 1a). We exposed 32 bacterial pathogens to 10 infection-relevant stress conditions. Also species-specific in vitro virulence inducing conditions were included if described in literature. As control for differential expression analyses, we utilized unexposed, exponentially grown bacteria (Fig. 1a, Table 1, Supplementary Data 1, and Supplementary Fig. 1a). We obtained an average of 12.2 million reads for each rRNA-depleted barcoded library, which far exceeds the 2–3 million reads considered sufficient to determine differentially expressed genes from bacteria with high significance[22] (Supplementary Data 2). Hierarchical clustering of expression profiles clustered replicates of the same sample together, indicating robust measurements of gene expression in all conditions and strains (Supplementary Fig. 1b). To asses quality of read mappings, also transcript length coverage was evaluated for each strain using the replicate with the lowest number of total reads. This showed accurate distribution of reads along the transcript length even in the poorest sequencing libraries, indicating good mapping quality (Supplementary Fig. 2).

Differential expression analysis showed that all bacteria dynamically responded to the stresses by regulating 62–90% of their genes in at least one condition (Fig. 1b). The highest fraction of regulated genes was measured under hypoxia, nutritional downshift, and stationary phase (Fig. 1c). Accuracy of the analyses was verified by differential expression of genes expected to be regulated under certain stress conditions (Supplementary

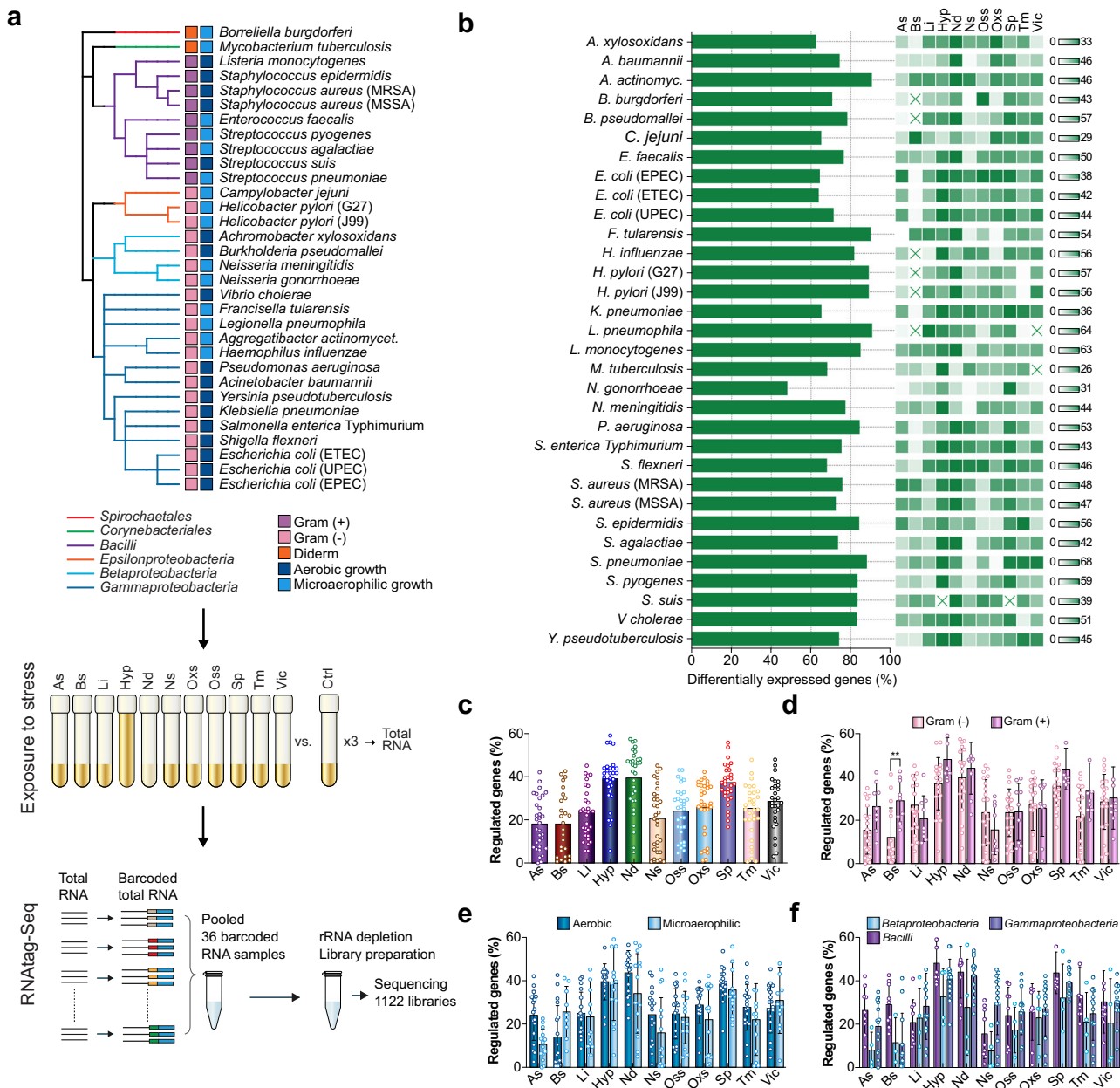

**Fig. 1 Stress responses of cross-microbial human pathogens. a** Phylogenetic clustering of bacterial species included in the study together with a schematic illustration of the experimental setup for 11 infection-related stress exposures (see also Table 1) and RNA-seq library preparation with RNAtag-Seq allowing multiple samples (36 in this study) per library used for obtaining 1122 transcriptomes deposited in the PATHOgenex RNA atlas. Phylogenetic orders, Gram staining groups, and oxygen dependency are indicated by color. **b** Proportion of genes differentially regulated in at least one of the conditions for each species. The linked heat maps show proportions of regulated genes for each stress condition. The scales to the right show the span from zero (white) to the highest (green) proportion (%) of regulated genes in the stress condition with the highest percentage of regulated genes. Heat map squares marked with a cross indicate that RNA-seq was not performed for that specific condition. **c** Dot plot showing the percentages of genes differentially regulated in each condition for all included species, **d** for Gram-negative and -positive bacteria, **e** for bacterial groups with aerobic and microaerophilic growth, **f** for three main phylogenetical orders included in this study. $n = 32$ species were examined for As, Li, Nd, Ns, Oss, and Oxs; $n = 26$ were examined for Bs; $n = 31$ were examined for Hyp; $n = 31$ were examined for Sp; $n = 30$ were examined for Vic in **c**. $n = 21$ Gram-negative and $n = 8$ Gram-positive; $n = 17$ aerobic and $n = 15$ microaerophilic bacterial species were examined in **d** and **e**, respectively. $n = 4$, $n = 14$, and $n = 8$ bacterial species from *Betaproteobacteria*, *Gammaprotobacteria*, and *Bacilli* were examined in **e**. Data are presented as mean values in **c** and as mean values ± SD in **d**, **e**, and **f**. The significance between the groups in **d**, **e**, and **f** was calculated with two-tailed Multiple t-test using Holm-Sidak method by Prism Graphpad version 8.2.0. ** indicates *p*-value = 0.0033. Source data are provided as a Source Data file.

Fig. 3). The *bfd* gene encoding bacterioferritin ferroxidase, shown to be upregulated under iron starvation[23,24], was upregulated under low iron condition in all strains harboring the gene. Similarly, *proW* encoding a permease that is part of the osmotically inducible ProU ABC transporter system[25], was upregulated in many of the tested strains under osmotic stress. Further, *dps* encoding DNA protection during starvation protein, shown to be positively regulated in an RpoS-dependent manner during stationary phase[26], was upregulated in many strains during stationary phase. The observed downregulation in *E. coli*

**Table 1 The consensus in vivo relevant stress conditions. The stress conditions were kept as similar as possible for the accuracy. Minor changes were done for certain species in certain conditions. Detailed information for the growth and stress conditions are shown in Supplementary Data 1.**

| Condition | Growth Medium | Growth Temperature | OD$_{600}$ | Stress exposure | Exposure Time |
|---|---|---|---|---|---|
| Control (Ctrl) | Ambient Med. | Ambient Temp. | 0.1–0.5 | — | 10 min |
| Acidic stress (As) | Ambient Med. | Ambient Temp. | 0.1–0.5 | pH: 3–5 | 10 min |
| Bile stress (Bs) | Ambient Med. | Ambient Temp. | 0.1–0.5 | 0.5% Bile Salts | 10 min |
| Low iron (Li) | Ambient Med. | Ambient Temp. | 0.1–0.5 | 250 µM 2,2-dipyridryl | 10 min |
| Hypoxia (Hyp) | Ambient Med. | Ambient Temp. | 0.1–0.5 | Low oxygen | 3–4 h |
| Nutritional downshift (Nd) | Ambient Med. | Ambient Temp. | 0.1–0.5 | 1X M9 salts | 30 min |
| Nitrosative stress (Ns) | Ambient Med. | Ambient Temp. | 0.1–0.5 | 250 µM Spermine NONOate | 10 min |
| Oxidative stress (Oxs) | Ambient Med. | Ambient Temp. | 0.1–0.5 | 0.5–10 mM H$_2$O$_2$ | 10 min |
| Osmotic stress (Oss) | Ambient Med. | Ambient Temp. | 0.1–0.5 | 0.5 M NaCl | 10 min |
| Stationary phase (Sp) | Ambient Med. | Ambient Temp. | 0.5–2.5 | Stationary phase | 3–16 h |
| Temperature (Tm) | Ambient Med. | Ambient Temp. | 0.1–0.5 | 41 °C | 20 min |
| Vir. ind. cond. (Vic) | Ambient Med. | Ambient Temp. | 0.1–0.5 | Varies | Varies |

(UPEC) and some other species such as *Streptococcus pyogenes* could be due to OxyR mediated upregulation in exponential phase[27]. As expected, *cstA*, encoding carbon starvation protein[28], was induced under nutritional downshift in the majority of strains harboring the gene. We also observed upregulation of *cstA* paralogues in strains harboring these. The observed upregulation of *ahpC*, encoding alkyl hydroperoxide reductase subunit during oxidative stress, is in accordance with what is known for many bacteria[29]. This gene was downregulated in *H. pylori* and not regulated for *F. tularensis* and *Haemophilus influenzae*, suggesting possible involvement of other hydrogen peroxidases. The gene encoding the heat shock protein GroL was upregulated in majority of the strains under temperature stress. Upregulation of *hmp*, encoding flavohaemoglobin known to be induced under nitrosative stress[30], was here observed in the majority of strains under this condition. Consequently, differential expression of genes indicative of responses to applied stresses are consistent with that observed by others, supporting accuracy of our differential gene expression analyses.

We also analyzed levels of responses for different groups of bacteria, such as Gram-negative and -positive, aerobic and microaerophilic, and different phylogenic orders (Fig. 1d, e, f). The fraction of regulated genes were found to be relatively similar between different groups. For Gram-negative and -positive strains, however, responses to bile were significantly higher in the latter group (Fig. 1d). Exceptions here were *Aggregatibacter actinomycetemcomitans*, *Campylobacter jejuni*, *F. tularensis,* and *Neisseria meningitidis* (Fig. 1b). The reason for this discrepancy in response to bile is not clear, but it is believed that Gram-positive bacteria are more sensitive to bile in general, and that the Gram-negative outer membrane with lipopolysaccharides contributes to bile protection[31], which might explain the lower response. However, bacterial species adapted to the mammalian intestine are often resistant to bile, and since the majority of intestinal bacteria in the strain collection are found in the Gram-negative group, we cannot exclude that this bias also is reflected in the comparison.

**Clustering genes into gene groups for cross microbial comparisons.** Comparing differential gene expression across different bacterial species is important for broad and comprehensive understanding of gene regulation and function in adaptation to new environments. Therefore, we clustered genes from the 32 strains into gene groups using two different orthology and homology approaches: one based on functional orthologs using KEGG's annotation tool GhostKOALA[32] for KO groups, and one

based on isofunctional homologs using the PATRIC database with Patric Global Family (PGFam)[33]. KO numbers are assigned genes harboring the same function, independent of sequence homology, while PGFam numbers are assigned genes with same function and sequence homology (isofunctional homologs) as indicated by RAST and CoreSEED[34]. PGFam groups are more specific in terms of function and homology, but with fewer genes per group. The tools assigned 6054 KO numbers to 63,831 genes, representing 60.7% of totally 105,088 genes in the dataset, and 27,999 PGFam numbers to 97,565 genes, representing 92.8% of the total number of genes (Fig. 2a).

**Revealing probability of gene groups to be differentially expressed.** To predict probability of genes within KO or PGFam groups to be regulated under certain conditions, we formulated an equation (Eq. 1) that computes a stress condition-specific score indicating 'probability to be differentially expressed (PTDEX)'. The equation was employed to gene groups with at least 2 genes (5340 KO groups and 11,353 PGFam groups) (Supplementary Data 3 and 4). The PTDEX score takes into account how conserved the genes in the group are among the 32 pathogens, what proportion of the genes are differentially regulated under a particular stress condition, how well this regulation is preserved among the species in the database, and what number of genes are regulated. We calculated stress condition-specific PTDEX scores for KO and PGFam groups as

$$\frac{n_{genes}(on)}{n_{genes}(total)} \cdot \frac{n_{species}(on)}{\sqrt{n_{strain}(total) \cdot n_{strain}(database)}} \cdot \log_2(1 + n_{genes}(on)) \quad (1)$$

The equation was developed in three parts, evaluating the different criteria mentioned above. In the first part, proportion of regulated genes in a particular gene group under a certain stress condition was calculated. This calculation gives a value regardless of large (well-conserved) or small (less-conserved) gene groups, hence gives no indication regarding how well this regulation is conserved among the species in the database. Therefore, in the second part, the frequency of this regulation among the species in the database was calculated. Here the number of observations at the species level, not strain level was calculated, as there are species with more than one strain in the database. This was done to ensure that a likely similar regulation of same genes in different strains of a species not affects the frequency. As bacterial strains in the database are diverse in terms of phylogenetical order and gene content, they are not expected to harbor genes from all gene groups. Therefore, frequency is calculated as number of species having at least one regulated gene divided by square root of

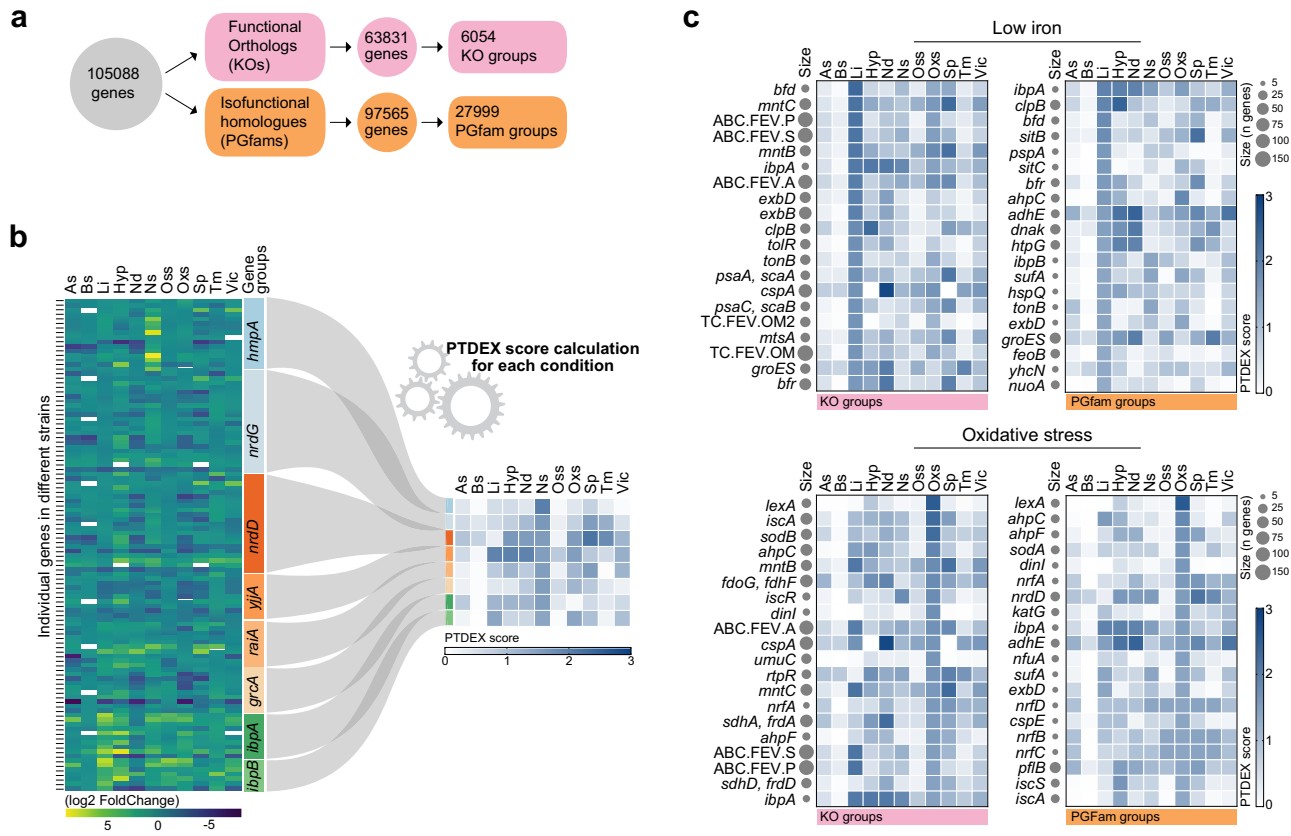

**Fig. 2 Transformation of differential expression to PTDEX scores for cross-microbial comparisons. a** Schematic illustration of clustering of 105,088 genes based on functional orthology (KO) and isofunctional homology (PGFam) and resulting KO and PGFam gene groups. **b** Illustration of the transformation of differential expression values of genes from many pathogens that are clustered in one gene group into a PTDEX score for each stress condition. Eight gene groups with highest PTDEX scores in nitrosative stress condition are shown as example. White rectangles indicate no differential expression. **c** Heat maps showing the 20 KO (left) and PGFam (right) groups with highest PTDEX scores in low iron (top) and oxidative stress (bottom) conditions. The size of gray dots after the gene name relates to the number of genes in the gene groups. Source data are provided as a Source Data file.

strains harboring the gene times strains in the database. The first two parts of the score provide relative values of proportion of regulated genes and frequency of their regulation, but not number of regulated genes, thus reflecting only conservation of gene groups and their regulation. Therefore, in the third part, the log2 value of total number of genes that are regulated in the gene groups was calculated and value of one was added to avoid obtaining a factor of zero for non-regulated genes. The log2 value was used to minimize the effect of well conserved gene groups against poorly conserved gene groups. This was very critical for PTDEX score equation to give higher scores to more particular stress associated gene groups then well-conserved genes unlike to other statistical analysis such as hypergeometric test. While the lowest PTDEX is zero due to no regulated genes, we observed the highest PTDEX score (2.83) for PGF_00016395 (*rpmD*) under hypoxia and stationary phase. This group has 29 genes encoded in 28 species of which 23 genes in 23 species are differentially regulated under hypoxia and stationary phase. The PTDEX score is a transformation of differential expression of individual genes from different species under each stress condition to a single value (Fig. 2b) that can be used for comprehensive understanding of gene regulation in bacteria.

As the calculation measures conservation of the genes and their regulation among tested bacteria, the PTDEX scores of gene groups that are highly conserved among the 32 strains but not regulated in many would have similar PTDEX score with gene groups that are less conserved but regulated in a high proportion of strains harboring the gene. For example; *leuS* (PGF_06812369)

is conserved in 31 strains and regulated in only 11 species under nutritional downshift and *gcvPA* (GF_00008774) is conserved in 6 strains and regulated in 4 species have similar PTDEX score (0.43 and 0.44, respectively) for nutritional downshift. Score values of ≥0.25, a value where at least 50% of the genes in the groups are differentially regulated regardless of the number of genes in the group was considered as 'high PTDEX score' (Supplementary Fig. 4a).

To test the power of PTDEX scores, we ranked KO and PGFam groups based on PTDEX scores in low iron and oxidative stress and plotted those representing the highest 20 (Fig. 2c). Many of the genes with high KO and PGFam PTDEX scores for low iron represented genes associated with iron uptake and iron homeostasis. Similarly, the genes with highest scores in oxidative stress were genes known to respond to DNA damage and oxidative stress (Fig. 2c). As expected, the highest 20 were not exactly the same for the KO and PGFam groupings, but some of the expected gene groups were found in both; such as *bfr, bfd, tonB*, and *exbD*[15,23,24] for low iron and *lexA, dinI, ahpC*, and *ahpF*[29] for oxidative stress (Fig. 2c). Furthermore, all the gene groups representing genes indicative for the different stress conditions shown in Supplementary Fig. 3a have high PTDEX scores (Supplementary Fig. 4b). Interestingly, we observed similar levels of PTDEX scores for KO and PGFam groups. Similar PTDEX score for relatively smaller PGFam groups likely reflects a more similar regulation of isofunctional homologs than of functional orthologs. Therefore, for more accurate conclusions, we used PGFam PTDEX scores for the remainder of our analyses.

**Comparisons of stress responses between evolutionary distinct bacterial groups**. We next re-clustered genes of Gram-negative and -positive strains into G- and G+ PGFam groups and re-computed the PTDEX scores separately (Supplementary Data 5 and 6). We then generated a similarity matrix with the Pearson correlation coefficient for PTDEX scores from 7105 G- PGFam groups, representing 47,468 genes, and 2903 G+ PGFam groups, representing 14,200 genes, for each stress condition. There were significant differences in responses by Gram-negative versus -positive strains, which supports previous studies[35] (Fig. 3a). Transcription factors, especially global regulators, are poorly conserved, and transcriptional regulation has been found to be more flexible than their target genes[36]. In support of this, an analysis of differential expression of transcription factors in response to the different stresses revealed high diversity between Gram-negative and -positive strains (Supplementary Fig. 5). However, despite substantial differences in PGFam groups, parts of the responses were shared by Gram-negative and -positive bacteria, where conditions showing the highest overlaps were hypoxia, nutritional downshift, and stationary phase (Fig. 3b). These analyses shows usage of PTDEX score as a novel approach providing in-depth overviews of common and specific regulations of the same genes in different species, simplifying cross microbial comparisons.

**Identification of intersections between stress responses**. Reponses to different stressors are expected to partly overlap due to involvement of the same molecular pathways and functional diversity of gene products. Many responses are expected to involve halted growth, enabling translational adjustments and redirection of resources. We next used PTDEX scores of G-/G+ PGFam groups to identify overlaps between different stress responses. There were higher degree of overlaps in Gram-negative compared to that of Gram-positive bacteria, where especially hypoxia, stationary phase, and nutritional downshift overlapped to high extent. Also responses to low iron, oxidative, and nitrosative stress, as well as responses to acidic and osmotic stress overlapped (Fig. 3c). To reveal overlapping pathways and processes, we implemented a co-expression module identification algorithm[37] on G- and G+ PGFam PTDEX scores. This generated six modules of PGFam groups for Gram-negatives showing common patterns of PTDEX scores of PGfam groups in certain combinations of stress conditions, and 3 modules in Gram positives. KEGG pathway mapping of genes in these modules indicated overlapping pathways and processes (Fig. 3d, Supplementary Data 5). The higher number of genes involved in overlapping pathways in N-module-1 and P-module-1 is indicative of more global responses to hypoxia, nutritional downshift and stationary phase. In accordance, genes encoding RpsA and RplA, involved in ribosome biogenesis and ArgH and LysC involved in amino acid biosynthesis were differentially regulated under those conditions in both Gram-negative and -positive strains (Fig. 3d). As seen in N-module-2, genes encoding ribonucleoside-diphosphate reductase subunits (*nrdA, nrdB*), and AMP nucleosidase (*amn*) involved in Purine-Pyrimidine metabolism were regulated in response to low iron, nitrosative stress and oxidative stress, conditions known to induce DNA damage[38]. Accordingly, *nrdA* and *nrdB* was previously shown to be induced upon exposure to DNA damaging agents in *E. coli*[39]. The dataset did not allow adequate comparisons between phylogenetical orders due to the uneven distribution of strains; only *Bacilli* representing Gram-positive strains (9 strains) and dominance of *Gammaproteobacteria* among Gram-negative strains (3 *Epsilonproteobacteria*, 4 *Betaproteobacteria*, 14 *Gammaproteobacteria*).

**Revealing universal stress responders including known antimicrobial targets**. Participation of gene products in responses to multiple stresses can be key to conservation during evolutionary diversification of species. To retrieve genes encoding these universal stress responders (USRs), we selected PGFam groups with high PTDEX scores (≥0.25) in at least six conditions and containing at least 11 genes from Gram-negative strains (21 in total) and 4 from Gram-positive strains (9 in total) (Supplementary Data 7). We identified 168 USRs (from 6465 individual genes), where functional clustering with PATRIC subsystems[40] revealed that USRs are involved in basic biological processes such as metabolism, energy generation, ribosome biogenesis, amino acid biosynthesis, cell division, RNA processing, membrane transport (Supplementary Fig. 6). Interestingly, 9 of the USR genes encode targets for antibiotics, where mutations in their sequences been shown to confer antibiotic resistance (Supplementary Fig. 6). Hence, novel putative antibiotic targets might be found among remaining 159 USRs. In accordance, *nrdD* and *nrdG* genes encoding Class III ribonucleotide reductase, display high PTDEX score in all conditions has indeed been suggested as a target for compounds to inhibit cell growth[41].

An attractive approach for future antimicrobials is the development of narrow-spectrum or even species–specific drugs to avoid cross-resistance in non-targeted bacteria, and we therefore also retrieved genes representing species-specific USRs (Supplementary Data 8). This group includes genes that are specific to a particular species in the PATHOgenex dataset and differentially regulated under at least 6 stress conditions. We identified 2194 species-specific USRs of which 960 were annotated as genes encoding hypothetical proteins, indicating that a considerable portion of species–specific stress response is still unknown and remain to be explored.

**Stress responses identified in PATHOgenex are relevant for infection in vivo**. To reveal the in vivo relevance of the data collection of in vitro obtained transcriptomes, we employed expression profiles for clinically relevant pathogens obtained during in vivo infection and compared these to PATHOgenex data. We utilized recently published transcriptomes of the Gram-negative *P. aeruginosa* in cystic fibrosis lungs[42], and the Gram-positive *S. aureus* (MSSA) in acute murine osteomyelitis[43]. The comparisons showed that the majority of (75–83%) of the in vivo regulated genes of both pathogens were co-regulated in at least one of the in vitro stress conditions (Fig. 4a). Notable, 3–12% of the regulated genes represented USRs, highlighting the potential importance of those genes for adaptation to real host environments. In vivo regulated genes not co-regulated in any of the in vitro conditions indicate existence of in vivo specific regulation. This analysis showed that mimicking host stress conditions is a relevant approach for studies aiming at understanding of stress dynamics linked different infection scenarios.

To reveal specific stress responses associated with these infections, the number of genes for each in vitro stress condition that also was regulated during infection was determined. Hypoxia, nutritional downshift, and stationary phase responses were excluded as these conditions are complex and involve multiple overlapping stress responses. This showed that responses to nitrosative stress and osmotic stress were pronounced during *P. aeruginosa* lung infection (Fig. 4b). *P. aureginosa* is known to be exposed to low iron, oxidative stress, and osmotic stress in microaenvironments of cystic fibrosis lungs[42,44,45]. However, this is mostly based on functional annotation analysis using Gene Ontology and KEGG pathway mapping. Although such analyses provide information of infection dynamics, they are based on very

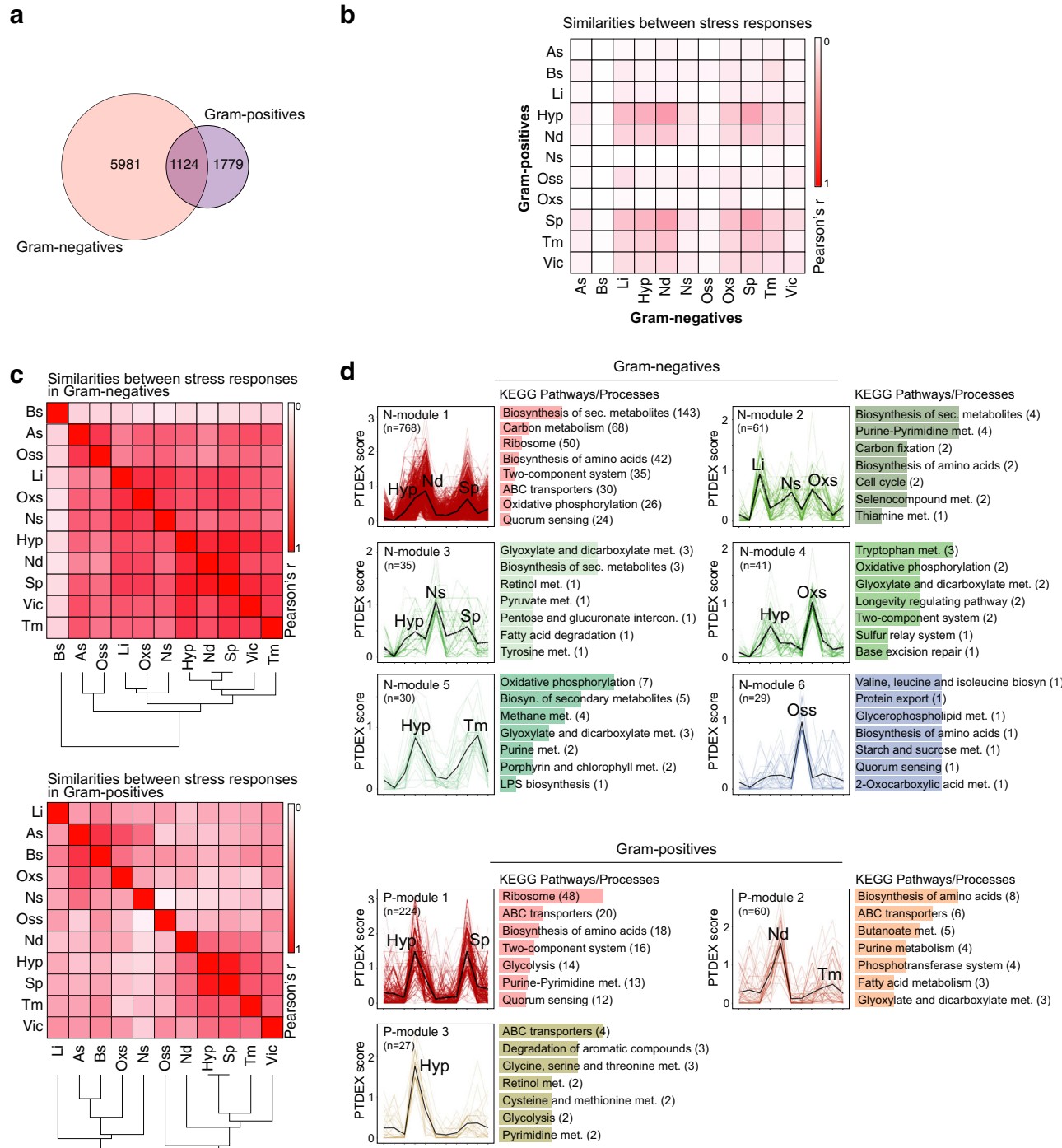

**Fig. 3 Gram-negative and -positive bacteria exhibit distinct responses.** **a** Venn diagram of individual and shared PGFam groups with a PTDEX score >0 in at least one of the stress conditions. Venn diagram was generated with http://bioinformatics.psb.ugent.be/webtools/Venn/. **b** Heat map showing degree of similarity between responses to different stresses in Gram-negative and -positive bacteria. The similarity was calculated using the Pearson correlation coefficient of PGFam groups PTDEX scores in each condition. **c** Heat map showing degree of similarity between responses to different stress conditions in Gram-negative and -positive bacteria. Similarities were calculated as in **b** and the similarity distances shown in a dendrogram. **d** Modules generated by CemiTool showing conditions that involve similar regulation of gene groups with high PTDEX scores in Gram-negative and -positive bacteria. *n* indicates number of PGFam groups within each module. The most enriched KEGG pathways/processes for each module are shown with the number of gene groups indicated in brackets. See also Supplementary Data 5 and 6.

low number of functionally annonated genes. Comparing to data in PATHOgenex, we could retrieve more comprehensive information about ongoing stress responses and found that response to nitrosative stress during *P. auroginosa* lung infections have been underestimated. The nitrosative stress related genes differentially regulated in infected lungs involve for example the

*narGHJI* operon and *narK* encoding proteins are involved in repair of damage caused by reactive nitrogen species[46–48]. Also *kguE, kguK, kguT*, and *kguD* genes encoding proteins involved in 2-ketogluconate biosynthesis and catabolism pathways[49] were induced during infection and nitrosative stress only. Response to osmotic stress was also rather pronounced during lung infection,

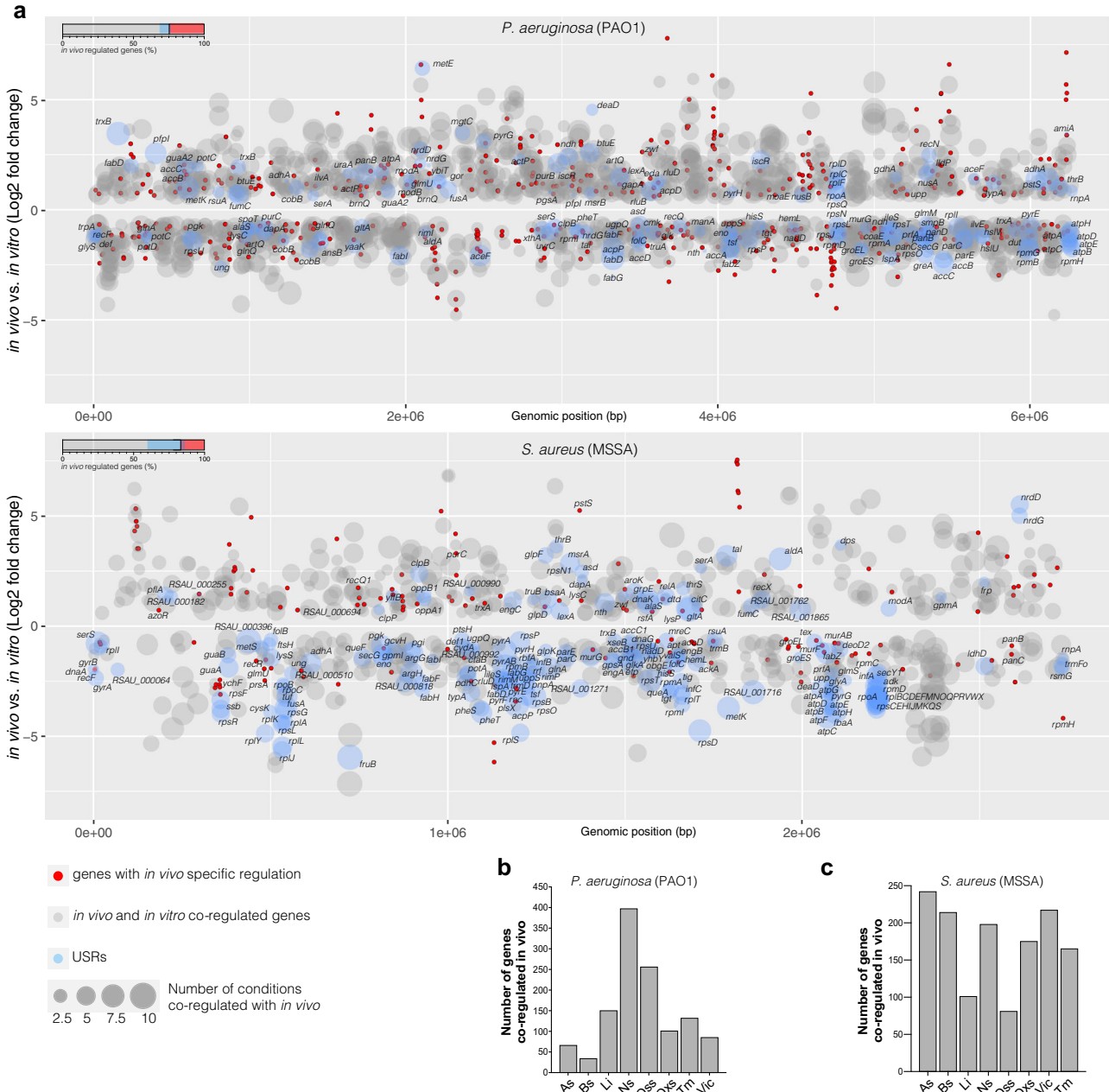

**Fig. 4 Stress responses identified in PATHOgenex are relevant for infection in vivo and can be used for determination of environmental stresses for pathogens at different infection niches. a** Differentially expressed genes obtained from differential expression analysis of in vivo transcriptomes of *P. aeruginosa* in cystic fibrosis lungs[45] and *S. aureus* (MSSA) during acute murine osteomyelitis[46] were mapped to in vitro stress responses of the corresponding species. Log2 fold changes in vivo vs. in vitro (control) are shown with genomic localization of genes for each species. The genes that are co-regulated (up/up or down/down) between in vivo and in vitro stress conditions are shown in gray or blue (conserved USRs) bubbles where the size of the bubble indicates number of conditions showing co-regulation with in vivo. Genes with in vivo specific regulation, which has no co-regulation at any of the stress conditions, are shown with red bubbles. The bars in the upper left corner of each bubble plot indicate the proportion of genes that are similarly regulated in vivo and in vitro (grey and blue, where blue indicate USRs) and proportion of genes showing in vivo specific regulation (red). **b** The number of genes that are co-regulated during infection for each of the PATHOgenex in vitro stress conditions for *P. aeruginosa* and, **c**, for *S. aureus*. Hypoxia, nutritional downshift, and stationary phase were not included due to the presence of variety of stress responses under those conditions. Source data are provided as a Source Data file.

involving expression of *pscU, pscF, osmC*, PA1323, and PA1324, previously shown to be induced under osmotic stress[50] and *betA, betB* genes encoding choline dehydrogenase and glycine betaine aldehyde dehydrogenase involved in osmoadaptation[51] (Supplementary Data 9). Similar analyses for *S. aureus* infection in the murine osteomyelitis model showed a different pattern, with many stress responses engaged, where low iron and osmotic stress

responses, were relatively lower than the others (Fig. 4c). The response to acidic stress was most pronounced and involved elevated expression of *purC, E and K*[52]. Osteomyelitis is associated with high level of inflammation and bone tissue destruction via osteoclastic resorption of bones[53]. Osteoclasts secrete hydrogen ions, collagenase, cathepsin K, and hydrolytic enzymes during acute infection where the hydrogen ions lead to

dissolution of bone minerals and an acidic miroenvironment, which was reflected in our comparison with pronounced acidic stress response (Fig. 4c). We also noted increased expression of a gene (RSAU_000352), encoding a super-antigen-like exotoxin during acute infection and in acidic stress only in vitro, suggesting a possible functional role for this toxin under acidic microenvironments during infection (Supplementary Data 9). The second most pronounced response during the acute *S. aureus* infection was measured for virulence inducing condition, which in this case was treatment of bacteria with human serum (Fig. 4c). Taken together, comparison of in vivo bacterial responses to stress responses in PATHOgenex could provide insight to environmental cues at in vivo infection sites and reveal adaptation strategies employed by bacterial pathogens.

**ncRNAs in bacterial stress responses.** Analysing overall transcription, we observed that a substantial proportion of RNAs were transcribed from non-coding DNA regions, including sRNAs, regulatory ncRNAs, 5′ untranslated regions (UTRs), 3′-UTRs, and intergenic regions (noteworthy, this did not include transcription of cis-antisense RNA encoded on CDS's opposite strand). The proportion is expected to be even higher than calculated, since probably many sRNAs were excluded during sample preparation due to the >100 nt cut-off for library preparation. In some bacteria, expression of non-CDSs was higher than that of CDSs in hypoxia, nutritional downshift, and stationary phase conditions. For example, 72% of reads mapped to *V. cholerae* genome were transcripts encoded from non-coding regions under hypoxia. Corresponding numbers for *E. faecalis* were 62% under hypoxia and 60% at stationary phase, and for *N. meningitidis* was 53% under nutritional downshift (Supplementary Data 2). Noteworthy, for most species, the percent of transcripts from non-CDS was increased upon stress exposure in comparison to control (Fig. 5a). Similar proportions of non-CDS transcripts and also stress-induced upregulations were also shown in a previous study of *Salmonella*[18].

The observed stress associated increases of transcripts from non-CDS in many cases correlated with increased expression of tmRNA, also known as SsrA, a conserved and abundant RNA in bacteria responsible to restore translation in detrimental situations[54]. Induction of tmRNA were particularly high at stationary phase, hypoxia, and upon nutritional downshift, which indeed are conditions associated with halted growth (Fig. 5b and Supplementary Data 10). In line with this, *smpB*, encoding a protein interacting with tmRNA, was identified as a USR with high PTDEX scores in those conditions. (Supplementary Fig. 6). The tmRNA level was however not correlated with expression ratios of non-CDS regions for all species (Fig. 5a, b). For example in *Klebsiella pneumoniae*, expression of the carbon storage regulatory ncRNA CsrB, involved in many biological processes[55], correlated with expression ratios of non-CDS regions in many stress conditions except stationary phase (Fig. 5a, d). For stationary phase we found high extent read mapping to 5′-UTR and 3′-UTR regions of a CDS (KPN_01149) encoding a hypothetical protein, which has 2 paralogs with diverse UTR sequences in the genome (Supplementary Fig. 7a, Fig. 5d, e). The expressed region started at a position reported to be a transcriptional start site[56], supporting the accuracy of this finding (Fig. 5e). Also *S. aureus* showed expression from non-coding regions not corresponding with expression of tmRNA. Here, high level of reads were mapped to SRS42, a 1232-nucleotide long ncRNA (Fig. 5f, g) previously shown to contribute to hemolysis and production of alpha-toxin[57]. Analysis of data from a previous study[43] showed that expression of SRS42 indeed is increased in *S. aureus* during infection (Supplementary Fig. 7b).

Another abundant ncRNA with potential to impact transcription widely spread among different bacteria is 6S RNA[58]. In *E coli*, 6 S RNA bind the $\sigma^{70}$ RNA polymerase and inhibit transcription, promoting adaption to stationary phase and environmental stresses[59]. Since expression profiles of many other bacteria suggest similar involvement of 6S RNA in adaption to stationary phase, but also potentially different roles in other[60], we analyzed of 6S RNA expression during different stresses in the PATHOgenex strains. The proportion of 6S RNA was here found to be rather stable under most stress conditions in comparison to control, but increased under hypoxia, stationary phase, and virulence inducing conditions in some species (Supplementary Fig. 8). The proportion of 6S RNA in *C. jejuni* and *M. tuberculosis* stood out among the other species with higher expression levels. 6S RNA in *Mycobacteria*, known as Ms1 RNA has been reported to have altered binding to the RNA polymerase, not requiring the holoenzyme[61], which might suggest an alternative role of this variant. However, also here the highest expression levels are seen for stationary phase and hypoxia, which is in line with that of many traditional 6S RNAs.

**The PATHOgenex RNA atlas provides opportunities for exploring gene expression in human pathogens.** All transcription data have been collected in an interactive database designed to serve researchers, the PATHOgenex RNA atlas (www.pathogenex.org). The data can be browsed by selecting strain(s) of interest and searched with one or multiple locus tags, protein ID, gene product, and PGFam groups. The database provides information of expression of pathways and operons across different stress conditions. As an example, we show that the three type VI secretion systems (T6SSs) operons in *P. aeruginosa* PAO1 are regulated under different stress conditions (Supplementary Fig. 9a). Such data can help in predicting operon structures, regulation of genes in operons under different stress conditions, or regulation of genes in specific genomic regions such as plasmids and genomic islands. PATHOgenex RNA atlas also allows the retrieval of most regulated gene groups under certain stress condition(s) in a wide range of bacterial pathogens. We show the top 5 highly regulated gene groups under virulence inducing conditions as an example (Supplementary Fig. 9b). The gene with highest PTDEX score under virulence condition was *adhE*, encoding bi-functional acetaldehyde-CoA dehydrogenase and alcohol dehydrogenase, previously shown to be important for *E. coli* and *S. enterica* virulence[62,63]. We could also see that *adhE* is differentially regulated in Gram-positive and -negative bacteria by retriving differential expression levels (Supplementary Fig. 9c). Moreover, the database can also provide additional information on gene expression level in individual species under all tested conditions (Supplementary Fig. 9d).

**Discussion**

Bacterial stress responses, which include coordinated regulation and interactions between different gene products involved in a variety of biological processes are critical for survival of pathogens and therefore attractive potential targets for future antimicrobials. However, genetically diverse pathogens harbor both orthologues and distinct biological processes to adapt to stresses[64], which hitherto has hampered investigations aiming at deciphering stress responses at global level. To meet this challenge, we have generated a high-quality resource with transcriptome datasets for comprehensive cross microbial analyses of orthologues and distinct biological processes. Comparisons of in vivo regulated genes of *S. aureus* and *P. aeruginosa* showed that the stress responses selected covered a majority of responses that occur during infection. The high quality of the PATHOgenex atlas rely on the

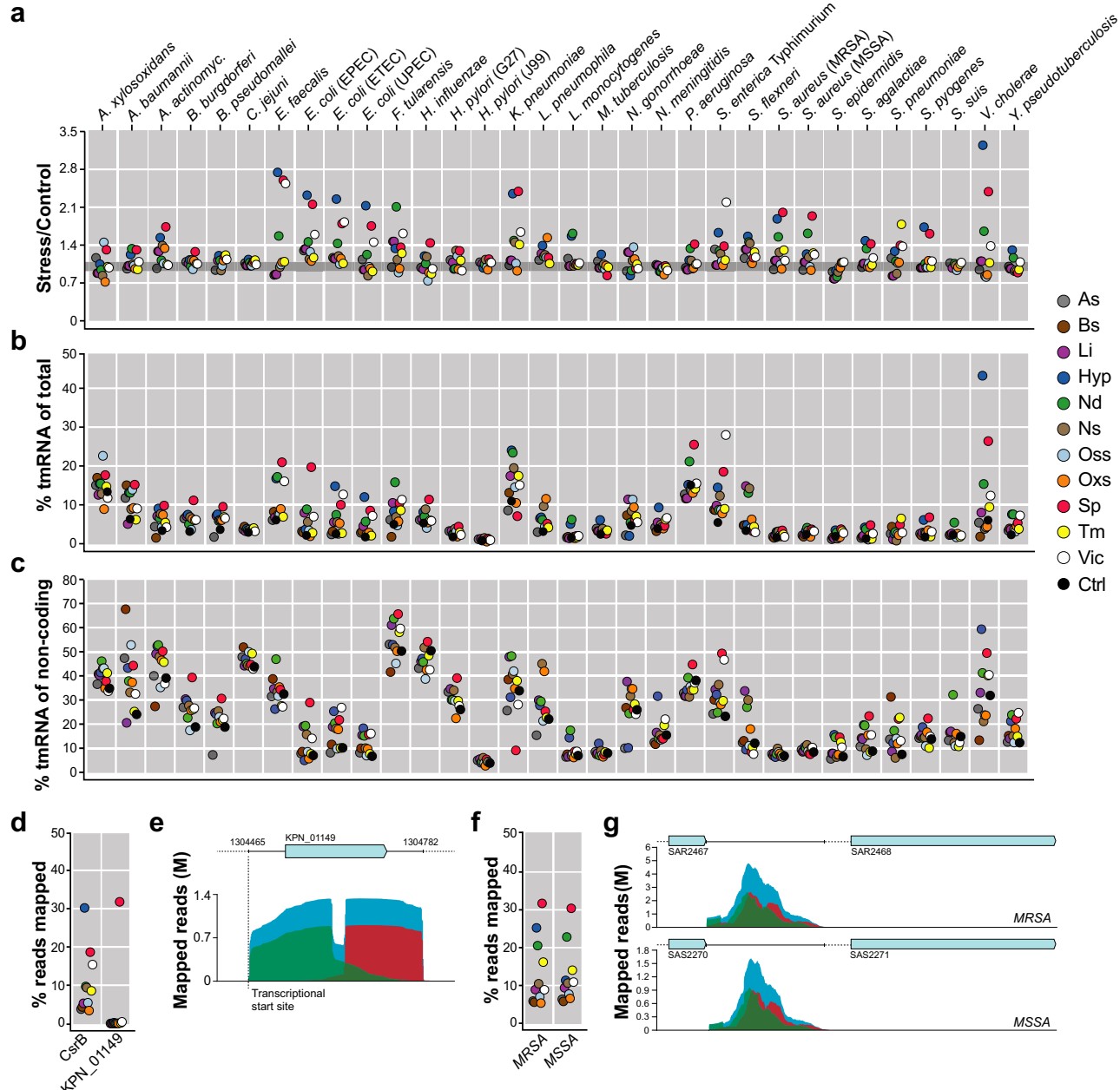

**Fig. 5 Complex stress conditions commonly involve differential expression from non-coding regions. a** Ratio of percent of reads mapped to non-CDSs under each stress condition in comparison to control for all tested strains. Different stress conditions are indicated by color (right). The gray line indicates Stress/Control value equal to one. **b** Proportion of reads mapped to tmRNA sequences of total number of mapped reads under each stress condition for all tested strains. **c** Proportion of reads mapped to tmRNA sequences of total number of reads mapped to non-CDSs under each stress condition for all tested strains. **d** Proportion of reads mapped to the *K. pneumoniae* CsrB and to the region covering KPN_01149 CDS with 5'-UTR and 3'-UTR sequences under each stress condition. **e** Number and alignments of reads mapped to KPN_01149 and its UTRs in stationary phase. Green indicates forward reads, red indicates reverse reads, and blue indicates paired reads. The vertical dashed line indicates the position of the TSS of KPN_01149[53]. **f** Proportion of reads mapped to SRS42 in *S. aureus* MRSA and MSSA genomes under each stress condition. **g** Number and alignments of reads mapped to SRS42 in *S. aureus* MRSA and MSSA strains in stationary phase. Colors represent reads as noted in **e**. Different stress conditions are indicated with colored dots on the right. The means of three replicates in each condition were used for the calculation in **a–d**, and **f**. Source data are provided as a Source Data file.

accuracy, depth, and dimension of the data. All experiments have been performed under similar conditions and resulting data have been handled similarly. This makes PATHOgenex a unique resource with high potential to provide novel information and also constitute a unique source for Big Data approaches. In contrast to challenging properties of existing data collections such as noisy, incomplete, or limited datasets originated from different labs with different experimental setups and bioinformatics

analyses[65–68], the PATHOgenex RNA atlas provides clean and complete datasets for machine learning algorithms aiming to understand complex biological problems in infection biology.

The high number and diversity of bacterial species in the PATHOgenex datasets allowed generation of a score system, providing response-specific PTDEX scores for iso-functional gene groups. These scores can be used to analyze the big dataset in terms of differential expression and regulation of common and

specific genes associated with certain responses in different bacterial groups. We here used the score to identify general USRs, a group of conserved genes involved in responses to multiple stresses in different bacteria, as well as species–specific USRs, which all can be interesting for exploring novel antimicrobial targets. The PTDEX score, which is indicative of regulation under particular stress condition(s), also provides hints about the function of products of genes encoding hypothetical and uncharacterized proteins. PATHOgenex can therefore be useful for designing experiments to reveal function of these gene products. This in turn can contribute with new information for functional annotations of microbial genes, which currently suffers from being poorly annotated, lacking information for a high proportion of genes. There is a clear need to increase the number of functionally annotated genes to facilitate biological understandings of transcriptomic data and the PATHOgenex RNA atlas can partly complement with expression levels and gene regulation data.

We also show examples where data in PATHOgenex can be used to get information of stress conditions encountered by pathogens during in vivo infections. Compared to the commonly used Gene Ontology and KEGG pathway mapping with a low number of functionally annotated genes, the use of PATHOgenex for comparative analyses increases possibilities to retrieve new information of bacterial microenvironments. We also found that significant proportions of transcriptional responses to stressful environments where from non-coding regions of the genomes. Besides known ncRNAs such as tmRNA and CsrB, we could show examples of ncRNAs that significantly increased in abundance during certain stresses in *K. pneumoniae* and *S. aureus*. Although not in the scope of this study, but given the high clinical relevance of these two pathogens, the potential role and function of these ncRNAs should be investigated further.

Taken together, we provide comprehensive information of gene expression at a single gene level in individual species, and at a broader level including gene groups, PTDEX scores and expression in multiple species. Results including different cross-microbial comparisons, retrieval of information about bacterial stress responses during infection, as well as the overall screen of transcripts from non-coding regions of bacteria exposed to environmental stresses shows the high potential of the PATHOgenex datasets as a data source. The dataset and associated findings have strength to give depth to comparative genomics studies and to more focused analyses of different pathogens. All together, PATHOgenex offers a rich source for researchers to generate novel hypotheses and design experiments accordingly, by that providing new opportunities for novel discoveries.

## Methods

**Bacterial growth and stress exposures**. All bacterial strains shown in Fig. 1a were grown in their optimal growth medium and temperatures in laboratories specialized in each species. The strength of the stress with the different agents and associated exposure time was designed to be as similar and relevant as possible for each species (Supplementary Data 1). Minor changes were necessary for certain conditions; for example, the "low pH level" used for *H. pylori* strains were lower than for the others. Three bacterial cultures were grown overnight and then subcultured to a new culture vial with 1:50–1:100 dilution. The cultures were grown until exponential phase with an $OD_{600}$ of 0.1–0.6 and exposed to the 11 stresses separately. Un-exposed cultures at exponential growth were used as controls for differential expression analysis. For nutritional downshift, bacterial cultures were spun down at 5,000 g for 2 min, the supernatant removed, the pellets resuspended in 1× M9 supplemented with 0.1 M $MgCl_2$ and 0.1 M $CaCl_2$, and incubated for 30 minutes at ambient temperature for each strain. For hypoxia, 2 ml screw cap tubes were filled with bacterial culture at exponential growth and the caps were screwed on without leaving space for air. The stress exposures were stopped by adding 0.5% (final concentration) phenol:ethanol solution (Supplementary Fig. 1a).

**Total RNA isolation**. One milliliter of triplicate bacterial cultures exposed to the stresses and un-exposed control culture were immediately pelleted by

centrifugation at $5000 \times g$ at room temperature for 2 min after adding phenol:ethanol. The supernatants were removed and pellets resuspended in 0.5 ml Trizol solution. For Gram-negative bacteria, cells were homogenized in Trizol solution by pipetting up and down 15 times. For Gram-positive bacteria, culture suspensions in Trizol were transferred to previously cooled bead beater tubes containing 0.1-mm glass beads and treated with Mini-Beadbeater (Biospec Products Inc, USA) twice at a fixed speed for 45 s, and then cooled on ice for 1 min between the treatments. Culture homogenates were incubated at room temperature for 5 min and then supplemented with 0.2 ml chloroform, thoroughly mixed by shaking 10 times, and incubated for 3 min. After centrifugation at $12,000 \times g$ at 4 °C for 15 min, the aqueous upper phase was carefully transferred to new RNase-free tubes. An equal volume of 99% ethanol was added to the aqueous phase and isolation continued using the Direct-Zol RNA Miniprep Plus (Zymo Research, USA) RNA purification kit protocol. Total RNAs were eluted in RNase free water in RNase free tubes. The total RNA concentrations were measured using the Qubit BR RNA Assay Kit (ThermoFisher Scientific, USA) and RNA integrity confirmed on a 0.8% agarose gel in TBE buffer.

**RNA-seq library preparation with RNAtag-Seq**. All rRNA-depleted RNA-seq library preparations were performed according to Shishkin et al. (2015), with minor modifications. RNAtag-Seq allows multiple library preparations in one tube, with initial tagging of total RNA samples with modified (5′P and 3′ SpcC3) DNA barcoded adaptors, each harboring a unique 8 nt sequence used to demultiplex individual libraries after sequencing. We combined the library preparation of three biological replicates from 11 stress-exposed samples and the un-exposed control sample, resulting in 36 library preparations in one tube for each bacterial strain. To tag the 36 replicates, 36 unique barcoded adaptors were used with every three barcodes used for samples from the same stress conditions in all bacterial species for consistentcy (Supplementary Data 2). A total of 100 ng total RNA was used for each biological replicate. The total RNA was fragmented in 2× FastAP Thermo-sensitive Alkaline Phosphatase buffer for 3 min at 94 °C, DNase treated, and dephosphorylated with a combination of TURBO DNase and Thermosensitive Alkaline Phosphatase in 1× FastAP buffer for 30 min at 37 °C. Fragmented, DNase-treated, and dephosphorylated total RNAs were cleaned with a 2× reaction volume of Agencourt RNAClean XP beads. Cleaned total RNAs were incubated with 100 pmol of the unique DNA barcode adaptors at 70 °C for 3 min, and then ligated with T4 RNA ligase 1 for 90 min at 22 °C. The ligation was stopped and the enzyme denatured by the addition of RLT buffer. The 36 denatured ligation mixes were then pooled and cleaned in a Zymo Clean & Concentrator™-5 column according to the manufacturer's 200 nt cut-off protocol. The RNAs were eluted in 32 µl of RNase-free water. Ribosomal RNA was depleted using the Ribo-Zero™ Magnetic Gold Kit (Bacteria) according to the manufacturer's instructions. The first-strand cDNA of each pool was generated using an AffinityScript Multiple Temperature cDNA synthesis kit with 50 pmol of AR2 primer at 55 °C for 55 min. The RNA was degraded by adding a 10% reaction volume of 1 N NaOH at 70 °C for 12 min and the reaction neutralized with an 18% reaction volume of 0.5 M acetic acid. After cleaning the reverse transcription primers with a 2× reaction volume of RNAClean XP beads, the 3Tr3 adapter was ligated with T4 RNA ligase 1 at 22 °C with overnight incubation. The second ligation was cleaned first with a 2×, and secondly 1.5×, reaction volume of RNAClean XP beads. The cDNA was then used as the template for PCR reaction with FailSafe™ PCR enzyme mix using 12.5 pmol 2P_univP5 as forward primers and 12.5 pmol ScriptSeq™ Index (barcode) PCR primers as reverse primers. The PCR cycles were as follows: 95 °C for 3 min, followed by 12 cycles at 95 °C for 30 s, 55 °C for 30 s, and 68 °C for 3 min, and then finishing at 68 °C for 7 min. The PCR product was cleaned first with a 1.5×, and secondly 0.8×, reaction volume of AMPure beads and eluted in RNAse free water. The library concentrations were measured using the Qubit™ dsDNA HS Assay Kit and the library insert size determined by the Agilent DNA 1000 Kit in a 2100 Electrophoresis Bioanalyser Instrument (Agilent, USA). The oligonucleotides used in RNAtag-seq sequencing are shown in Supplementary Table 1.

**RNA-seq data analysis**. The RNA-seq libraries generated by RNAtag-Seq were sequenced by either single-end or paired-end Illumina sequencing at SciLifeLab, Stockholm. Each library harboring a pool of 36 RNA samples was demultiplexed according to the unique 8 nt on the ligated barcode adaptors to separate reads from each replicate. The number of reads for the different barcoded samples did not reveal significant variation across conditions and species, excluding bias regarding over-representation of one or a set of barcodes during library preparation. The sequencing reads generated in this study were deposited in GEO with accession number GSE152295. Demultiplexed reads were then mapped to the genome of the species to which the RNAs belong in an annotation-independent manner. The accession numbers of the RefSeq reference genomes used for each strain can be found in Supplementary Data 1. The number of reads mapped to CDSs, rRNA, and tRNA was calculated according to the annotation files. Read mapping to reference genomes showed efficient rRNA depletion, with only 0.1–5% reads mapping to rRNA and tRNAs, except for *Campylobacter jejuni*, two of the *H. pylori* strains, and *Neisseria gonorrhoeae*, which had higher proportions mapping to rRNA (Supplementary Data 2). The library sequencing for these species was deep enough to cover 94% of the coding sequences (CDSs), with >10 reads per CDS in the least efficiently depleted library. The number of reads mapped to the non-coding regions

were calculated by subtracting the reads mapped to the annotated regions from the total number of mapped reads. The sequencing reads from primary transcripts of *K. pneumoniae*[56] with accession number SRR408498 were downloaded from Sequence Read Archive (SRA) and mapped to the same reference genome used in this study. In vivo and in vitro RNA-seq reads of *S. aureus* strain 6850 (accession number ERP005459)[43] and *P. aeruginosa* PAO1 (accession number GSE119356)[42] were downloaded from GEO and mapped to RefSeq reference genome CP006706 and NC_002516, respectively. The mappings of in vivo originated reads were performed with at least 95% correct matching to discriminate reference genome-specific reads and avoid mapping of host transcripts. Reads from *P. aeruginosa* lung lower peripheral lobe sample were excluded due to low transcriptome coverage[42]. All of the demultiplexing and read mapping steps were performed in CLC Genomic Workbench (Qiagen, USA).

For differential gene expression analysis, the trimmed mean of M values (TMM)[69] normalization method was used to normalize the sequencing depth of the individual libraries. For each bacterial species, the comparisons were performed between multiple groups and the control sample so that the full dataset could be used for fitting the generalized linear model. Using the complete datasets with multiple group comparisons allowed us to determine whether a gene has unstable expression for which the variation is random or is differentially expressed in response to the stress. Read values for genes with a maximum group mean expression (the maximum average TPM value in the statistical comparison group) <20 were removed and a threshold of 1.5-fold change with FDR-adjusted *p*-value <0.05 was employed to determine differential expression.

**Comparisons of in vivo differentially regulated genes of *S. aureus* and *P. aeruginosa* to in vitro stress responses.** For comparison between the *S. aureus* strain 6850 strain used for the mouse infection study and in vitro induced stress responses of *S. aureus* MSSA A476 in PATHOgenex, homologs genes were identified with PATRIC's Proteome Comparison Service. For *P. aeruginosa* PAO1 the same strain was used as reference genome. Co-regulated genes in vivo and each in vitro stress conditions were determined with the same differential expression profiles such as down-regulated in both conditions or up-regulated in both conditions.

**Generation of a phylogenetic tree of 32 strains.** A phylogenetic tree of the 32 strains based on NCBI taxonomy was generated with PhyloT in Newick format. The visualization of the tree was achieved using iTol.

**Clustering 32 strains based on expression profiles.** Hierarchical clustering, with Euclidian distance, of 32 strains based on the percent of genes expressed (TPM ≥ 10) in all 12 conditions, percent of genes expressed in at least one condition, and percent of genes not expressed in any of the conditions was performed with ClustVis[70,71].

**Clustering 105,088 genes with orthology/homology.** The KEGG orthology groups were assigned for clustering genes based on functional orthologs. The amino acid sequences of all CDSs for each strain were uploaded to GhostKOALA[32] to assign the best KO group in the genus_prokaryotes KEGG GENES database. Clustering based on isofunctional homologs was performed with PATRIC's Proteome Comparison Service. The amino acid sequences of all CDSs for each strain were compared using its own annotated genome, if available, or the closest strain to assign the best PGFam group for each CDS.

**PTDEX score calculation.** The PTDEX score was calculated for the orthology/homology groups using Python scripts and Jupyter notebooks, relying on the open-source libraries NumPy and Pandas. The PTDEX score formula relies on the definition of differentially expressed genes given in "RNA-Seq Analysis" above and the equation shown in Eq. 1, where $n_{genes}(on)$ is the number of differentially expressed genes, maximum group mean >= 20, fold change >1.75 with FDR-adjusted *p*-value <0.05, in the orthology/homology gene group, $n_{genes}(total)$ is the total number of genes in the group, $n_{species}(on)$ is the number of species for which at least one of the orthology group genes is differentially regulated, $n_{strains}(total)$ is the total number of strains for which at least one of the orthology group genes is present, and $n_{strains}(database)$ is the total number of strains in the database ($n_{strains}(database)$ =32 for general PTDEX scores, 21 for Gram negative-specific, 9 for Gram-positive-specific PTDEX scores). The additional logarithmic component is a weight factor to give more relative importance to orthology/homology groups with more genes.

**Generation of co-expression modules and KEGG pathway enrichments.** The co-expression Modules Identification tool (CEMiTool)[37] was implemented on PGFam groups of Gram-negative and -positive strains to find PGFam groups with similar PTDEX score patterns over the 11 stress conditions. PGFam groups associated with each module can be found in Supplementary Data 5 and 6. Each PGFam group associated with the particular module, if possible, was converted in KO groups and KO groups were used in the KEGG orthology database to map pathways.

**PATHOgenex website construction.** PATHOgenex was built using PHP Laravel Framework 6.18.2 and PHP 7.4.4 for server-side data processing, Javascript ECMAScript 2015 for the front end, and D3.js 5.16.0 and Plotly 1.40.0 libraries for the generation of the interactive visualizations. Linux distribution CentOS-8 with the 64-bit kernel 4.18.0 running on four processor cores and 64 Gb of RAM is used to host the web service on the in-house computational cluster.

**Reporting summary.** Further information on research design is available in the Nature Research Reporting Summary linked to this article.

## Data availability
The datasets with sequencing reads and processed data generated/analyzed during this study are available at GEO under accession number GSE152295. Previously published *P. aeruginosa* RNA-seq data from lung tissue or pure cultures and *S. aureus* RNA-seq data from mouse osteomyelitis model and pure culture are available at GEO under accession number GSE119356 and at ENA under the accession number PRJEB6003, respectively. The PATHOgenex RNA atlas with global expression profiles of the 32 bacterial pathogens under 11 stress conditions and an un-exposed control condition as well as PTDEX scores of PGFam gene groups are publicly available at www.pathogenex. org. Source data are provided with this paper.

## Code availability
The custom scripts employed to calculate PTDEX scores from differential expression analysis data are available at Zenodo (https://zenodo.org/record/4708491#. YID1nOaxVE5).

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

## Acknowledgements

We are grateful to all the expert labs for opening their lab to us and help with bacterial culturing and stress exposure experiments. We thank Drs. Peter Lind, Teresa Frisan, and Saskia Erttmann for critical reading of the manuscript. The work was supported by Knut and Alice Wallenberg Foundation (No. 2016.0063), Swedish Research Council (No. 2018-02855), and Insamlingsstiftelsen, Medical Faculty at Umeå University to M

Fallman; Novo Nordisk Foundation (K. Avican was partly supported by Grant No. NNF17OC0026486, awarded to Dr. Emmanuelle Charpentier at MIMS, The Laboratory for Molecular Infection Medicine Sweden); ERC (Starting grant, No. 716063) to J. Tang; Academy of Finland Research Fellow Grant (No. 317680) to J. Aldahdooh.

## Author contributions

K.A. and M.F. conceived and supervised the project, and wrote the manuscript with input from J.A, M.T., F.M., J.T., K.B., and M.R. K.A. performed the experiments and analyzed the data. M.T., K.B., K.A, and M.F. designed and M.T. calculated PTDEX scores. J.A., J.T., K.A., and M.F. designed and J.A. constructed the PATHOgenex RNA atlas and webpage.

## Funding

## Competing interests

The authors declare no competing interests.
