## [Peer Review File · Nature Communications]

REVIEWER COMMENTS

Reviewer #1 (Remarks to the Author):

The manuscript "RNA Atlas of Human Bacterial Pathogens Uncovers Stress Dynamics Linked to Infection" by Avican et al develop a method to enable the comparison and cross analysis of gene expression patterns from experiments with different microorganisms and conditions. As a proof of concept, they performed a comprehensive RNAseq study where they compared the transcriptional stress response of 32 different human pathogens under 11 different in vitro stress condition, relevant for infection, and when possible an in vitro condition mimicking virulence inducing conditions, when possible (Fig 1, suppl. Fig 1, Table 1 & supp Table 1). They analyzed the functional orthologs with two different annotation strategies /databases KO (from KEGG) and PGFam (from PATRIC) (Fig 2) and developed a "PTDEX score" to be able to compare the gene expression in the different data sets between the genes of the evolutionary sometimes distant genomes under the 11 different conditions (Fig 2). After establishing this PTDEX score for their comparative analysis the authors looked more specifically into the analysis of distinct stress response strategies within and between Gram-negative and Gram-positive species observed in their RNAseq experiment (Fig 3, suppl Fig 2 & 3). The identification of "universal stress responders" as possible more general key elements during that analysis is a very interesting concept. Especially since some of them are known antibiotic targets and many more could be tested as target for antibiotics (suppl Fig 3, Fig 4) Then the authors went a step further and successfully compared their comprehensive in vitro data set with in vivo transcriptome RNA-seq analysis data sets generated in vivo during an infection (*Pseudomonas aeruginosa* in cystic fibrosis lungs (Häussler and colleagues) and *Staphylococcus aureus* during Osteomyelitis (Medina and colleagues)) (Fig 4). The authors also observed that non-coding sRNA including the more specialized tmRNA are involved in various stress responses (Fig 5).

This is a very interesting and I also believe important study, which would help to compare and analyze the transcriptional changes under all kind of perturbations and for different organisms. This method will also help to explore and maybe clarify the intricate connection between the more general stress response systems and the specific adaptation processes facilitating pathogenicity of bacterial cells.

Comments and suggestions:

-The text of the paper is quite short and brief and describes their results often only in a very general manner. The figures are quite complex, which probably reflects the very comprehensive and complex data set and I acknowledge that the figure legends are informative. Nevertheless, the text very often only refers to the Figures without further explanation. This can be a challenge when trying to understand the possible key results and at least highlighting important aspects or observations would be beneficial to the reader. The text presenting the results for the very complex Fig 4 is such an example and in the last paragraph of the results section the authors only briefly mention as an example the stress regulation of T6SS in *Pseudomonas* displayed in suppl Fig 5 not mentioning the other results depicted in the same suppl Figure.

-Since the title of the paper states that "Stress Dynamics Linked to Infection" are uncovered in this paper, I would actually like to read a bit more on this topic. Which cellular stresses and stress responses are actually involved during infections by what pathogens? Is there a common theme or key results from the data presented in Fig 4? What is the possible role of the different types of identified "universal stress responders" for pathogenicity? In my opinion not only the general but also a more informative description of at least some interesting examples observed in this study could be warranted.

-How strong is the impact of the different applied stresses on the viability and survival of the different investigated microorganisms (Fig 1, suppl. Fig 1, Table 1 & supp Table 1)? Are the chosen conditions (suppl Table 1), on one hand strong enough that the stress impact is large enough to be detected, but on the other hand not too toxic? To assess this, it might be informative to show the growth curves and/or comparing the colony

forming units before and after applying the respective stress conditions for these different experiments and microorganisms.

-For the experiments shown in Fig 3 the authors analyze their data based on the classic distinction by Gram staining. However, this classification might not be the best in reflecting their evolutionary grouping especially for the Gram-negative bacteria. In Fig 1 the authors already group the investigated bacteria by phylogeny. Wouldn't it be better to rather compare their transcriptional response inside, and between the phyla as established in Fig 1? Or would following such a more evolution-based classification with such a relatively low number of investigated organisms make no difference to the presented analysis and results classified by Gram positive/negative?

-The described results about the non-coding sRNA appear a little confusing to me. First the authors note "a substantial proportion of RNAs were transcribed from non-coding DNA regions" and they go on that this is even an underestimation. But how abundant are they really? And more importantly there are all kind of RNA species and many reasons why they are encoded in these regions.

The presence of the more well-known and conserved different types of sRNA's such as csrB interacting and interfering with regulatory proteins such as CsrA as observed in the manuscript is no surprise. And the tmRNA, prominent in Fig 5, is a specific small RNA already known to be important for stress response, since they are crucial for ribosomal quality control allowing to rescue stalled ribosomes and subsequent degradation of the not fully translated proteins in all bacteria. Another interesting sRNA interacting with a protein is the 6SRNA. Is this one detected in the data sets?

As the authors mention themselves, there are many possible roles that these various different types small-regulatory RNA's can play. They can act on many targets in trans or as antisense RNA's in cis. For some microorganisms these asRNA species were also reported to be much more abundant than estimated before. Sometimes sRNA's can also encode for small proteins, as the authors also suggest/observe.

A question would be, whether these small RNA's can be classified and analyzed comparable to the genes encoding proteins in this study?

Maybe a more in depths analysis of cis and trans acting small RNA, if possible with this approach, could then be a performed in a separate story.

Reviewer #2 (Remarks to the Author):

In an effort to provide a complete and consistent dataset of RNA differentially expressed during stress in pathogenic bacteria, the authors performed transcriptomic analysis of 32 human bacterial pathogens in 11 host related stress conditions using the RNAtag-Seq technique (Shishkin et al, Nat Methods. 2015 Apr; 12(4): 323–325). Appreciably, the bacterial cultures were performed in several laboratories with expertise for each pathogen. They produced an extensive amount of data (1122 transcriptomes in total) which will be made publicly available on the GEO database. They compared gene expression in the different stress conditions and validated their data doing comparison with expression profiles of genes previously known to be regulated under those stresses. In particular, they developed a probability score (PTDEX score) to compare the stress responses between bacteria. To validate the fact that the genes regulated in the stress conditions are relevant during infection, they compared their data with published in vivo datasets from *S. aureus* and *K. pneumoniae*. Finally, they identified genes with altered expression in several stress conditions, which they named "universal stress responders", and proposed that those genes could serve as antimicrobial targets. All the differential expression data in a publicly available database "PATHOgenex", which is easy to understand and use. This database will be helpful for all scientist working on pathogenic bacteria, allowing to check differential gene expression in many stress conditions.

In my opinion the study is clear, well written and the conclusions are supported by the data, however I have some comments regarding the manuscript and suggestions for the database. In general, I would appreciate to know more about the elaboration of the PTDEX score and to have more comments and references about previously published work. Moreover, the paragraph focusing on non-coding RNA could be clarified.

Comments:

- Line 78: More comments about Suppl Fig 2a would be welcome in the text e.g. How were these genes selected among known stress regulated genes? Could the authors add references validating the function of those genes in stress response? Was it expected that given genes could be up or down regulated depending on the bacteria? In the legend please define what is the meaning of the crossed squares.

- Line 83: The authors found that response to bile is significantly higher in G+ bacteria than in G-, is it an overall observation? Or is the variation in only few G+ strains responsible for this difference?

- Line 107 and Fig 2b: The authors should elaborate more on the equation – in the text or M&M – explaining precisely how it was derive, if other scores where tested and why this one was chosen. In addition, after explaining the PTDEX score the authors should give detailed information about the meaning of the score. For instance, what do the lowest and highest values of the score correspond to? They could also give several examples describing different range of scores (or maybe describing how one can obtain the same score for different reason such as maybe important DE with few genes vs moderate DE with many genes). It is written line 153 that when the PTDEX is greater or equal to 0.25 at least 50% of the genes in the group are differently expressed. This information could already be mentioned there to give the reader a more precise idea about the meaning of the score. In addition, the PTDEX score corresponding to the DE expression shown in Suppl fig 2 could be presented.

- Figure 2b and Line 390: In Fig 2b, one parameter of the equation is nspecies(on) while in the M&M it is nstrain(on).

- Line 113 and Fig 2d: The authors mention “the expected genes groups that are found in both KO and PGFam groupings”, what are those genes? Could the authors describe their function add relevant references? The overlapping genes should be precisely identified in the figure.

- Fig 3b and 3c: The titles of the heat maps should be revised, both have now the same title while one is comparing G+ and G- responses (for each stress), and the other is comparing the stress responses (in G+ and G-).

- Line 144: Could the authors comment on the overlapping pathways and processes indicated by the KEGG pathways? Giving hints about the relation between the pathways and some examples.

- Suppl Fig 3: It would be useful to also define USR in Fig. legend.

- Paragraph starting at Line 147: A total of 421 USR with a PTDEX score greater or equal to 0.25 were retrieved, but how many different genes where considered in total for this analysis? It seems that only 609 PDFams groups were selected for this analysis, therefore it is expected to find genes involved in basic biological processes and this should be discussed in the text. Maybe it would also be of interest to find the top stress responders specific for each strain, those could pinpoint to strain specific putative antimicrobial targets.

- Line 180: Could the authors compare the proportion of non-CDS mapped reads with previously published data? The fact that a large fraction of transcripts originates from non-CDS has been published previously, as many publications report massive antisense transcription in bacterial pathogens.

- Line 191: The sentence is not clear. The authors state that the level of tmRNA is not high for all strains, for example *K. pneumoniae*: however, according to Fig. 4b, in this strain the percentage of tmRNA compared to total is one of the highest one (including in the control condition). The end of this paragraph is unclear, as well as the title of Suppl. Fig. 4.

Did the authors check that the intergenic region whose expression is increased during infection corresponded to an already published sRNA? (*S. aureus* database: <http://srd.genouest.org/>).

- Suppl Table 2: Could the authors comment on the fact that for some samples the percentage of mapped reads dropped below 50%. Do they think that those samples have a quality comparable to that of the other samples?

As the authors pinpointed, many regulated transcripts correspond to non-coding RNAs. It would be valuable to add data for the published small RNAs in the PATHOgenex database.

In the PATHOgenex database one condition is “Mig” while it is “Hyp” in the manuscript.

Did the authors considered the possibility to also give access to read coverage in a genome browser? If not, they should give example of read coverage for some genes per bacteria so that one can visualize sequencing data and assess the quality. In addition, it would be relevant to discuss already existing databases (transcriptomic data of human pathogens available in RNA

sequencing genome browsers): e.g. *H. pylori* (<http://hpylori-tss.imib-zinf.net/>), *Salmonella* (http://bioinf.gen.tcd.ie/cgi-bin/jay/salcom_v2.pl?_HL, <https://salmonella.wadsworth.org/>) etc.

Reviewer #1 (Remarks to the Author):

The manuscript "RNA Atlas of Human Bacterial Pathogens Uncovers Stress Dynamics Linked to Infection" by Avican et al develop a method to enable the comparison and cross analysis of gene expression patterns from experiments with different microorganisms and conditions.

As a proof of concept, they performed a comprehensive RNAseq study where they compared the transcriptional stress response of 32 different human pathogens under 11 different in vitro stress condition, relevant for infection, and when possible an in vitro condition mimicking virulence inducing conditions, when possible (Fig 1, suppl. Fig 1, Table 1 & supp Table 1).

They analyzed the functional orthologs with two different annotation strategies /databases KO (from KEGG) and PGFam (from PATRIC) (Fig 2) and developed a "PTDEX score" to be able to compare the gene expression in the different data sets between the genes of the evolutionary sometimes distant genomes under the 11 different conditions (Fig 2).

After establishing this PTDEX score for their comparative analysis the authors looked more specifically into the analysis of distinct stress response strategies within and between Gram-negative and Gram-positive species observed in their RNAseq experiment (Fig 3, suppl Fig 2 & 3). The identification of "universal stress responders" as possible more general key elements during that analysis is a very interesting concept. Especially since some of them are known antibiotic targets and many more could be tested as target for antibiotics (suppl Fig 3, Fig 4)

Then the authors went a step further and successfully compared their comprehensive in vitro data set with in vivo transcriptome RNA-seq analysis data sets generated in vivo during an infection (Pseudomonas aeruginosa in cystic fibrosis lungs (Häussler and colleagues) and Staphylococcus aureus during Osteomyelitis (Medina and colleagues)) (Fig 4).

The authors also observed that non-coding sRNA including the more specialized tmRNA are involved in various stress responses (Fig 5).

This is a very interesting and I also believe important study, which would help to compare and analyze the transcriptional changes under all kind of perturbations and for different organisms. This method will also help to explore and maybe clarify the intricate connection between the more general stress response systems and the specific adaptation processes facilitating pathogenicity of bacterial cells.

Please note that changes made according to reviewers' comments and suggestions in the revised version of the manuscript are indicated in, yellow-marked, in supplementary marked-up manuscript in supplementary material.

Comments and suggestions:

We appreciate the thorough review by this reviewer that wanted us to emphasize interpretation of the data on biological concepts and pathogenicity of bacteria. This reviewer provided many helpful comments and suggestions specifically for revealing percentage of 6S RNA under different stress conditions. We believe that addressing the comments and suggestions has greatly improved the revised manuscript.

-The text of the paper is quite short and brief and describes their results often only in a very general manner. The figures are quite complex, which probably reflects the very comprehensive and complex data set and I acknowledge that the figure legends are informative. Nevertheless, the text very often only refers to the Figures without further explanation. This can be a challenge when trying to understand the possible key results and at least highlighting important aspects or observations would be beneficial to the reader. The text presenting the results for the very complex Fig 4 is such an example and in the last paragraph of the results section the authors only briefly mention as an example the stress regulation of T6SS in Pseudomonas displayed in suppl Fig 5 not mentioning the other results depicted in the same suppl Figure.

We agree on this point. We have now expanded the text accordingly. In addition to expansion on parts recommended in other comments by this and by the other reviewer, text referring figures are now more informative about observations and when relevant possible importance of the result is suggested.

Fig 4 has been expanded with panel b and c and supported with a new supplementary table. The outcomes from Figure 4 are now highlighted and discussed (line 287-320, page 10-11, Supplementary Table 9).

For Supplementary Fig. 5; all panels are now mentioned and discussed in the text (line 374-385, page 13).

-Since the title of the paper states that "Stress Dynamics Linked to Infection" are uncovered in this

paper, I would actually like to read a bit more on this topic. Which cellular stresses and stress responses are actually involved during infections by what pathogens?

We have now expanded this part of the introduction and included additional information and examples of stress responses of different pathogens against different environmental cues at different human infection sites (line 34-65, page 1-2). We are aware of that this could be expanded even more, but is restricted by current limitations of word count and references.

Is there a common theme or key results from the data presented in Fig 4?

*This is a point that we are happy address and expand with additional information and results from new analyses. We have performed a more detailed analysis on the comparison of *Pseudomonas aeruginosa* and *Staphylococcus aureus* in vivo differentially expressed genes and in vitro regulation under the different stress conditions. The resulting data are presented in 2 additional panels of Figure 4, new panel 4b and 4c. These panels show the degree of the responses seen in vitro for the different environmental conditions used for the PATHOgenex database that is regulated in a similar way during in vivo infection, so called co-regulated genes for each condition. Additionally, we provide new information of specific genes and operons that are specifically co-regulated and associated to the pronounced stress responses seen in vivo (line 287-320 on pages 10-11). We have also included a new Supplementary table 9 with information of in vivo differentially regulated genes co-regulated in in vivo infection conditions. We find that these data further support and strengthen the relevance and potential of the PATHOgenex database.*

What is the possible role of the different types of identified "universal stress responders" for pathogenicity? In my opinion not only the general but also a more informative description of at least some interesting examples observed in this study could be warranted.

*We have now included extra information of some of the USRs, including *nrdD* and *nrdG* genes (line 263-266, page 9) and also a new section providing information about species-specific USRs (lines 267-274, page 9-10 and new Supplementary Table 8)*

How strong is the impact of the different applied stresses on the viability and survival of the different investigated microorganisms (Fig 1, suppl. Fig 1, Table 1 & suppl Table 1)? Are the chosen conditions (suppl Table 1), on one hand strong enough that the stress impact is large enough to be detected, but on the other hand not too toxic? To assess this, it might be informative to show the growth curves and/or comparing the colony forming units before and after applying the respective stress conditions for these different experiments and microorganisms.

This is an aspect that we thought about during planning of the experiments. On one hand we strived for as similar conditions as possible including concentration of stress agents and exposure time for the different pathogens included in the database. But on the other hand, we realized that using only generalized conditions for all bacteria, will not be informative enough for some of the species. Therefore, concentrations of some stress agents were adjusted for some species, as shown in Supplementary Table 1. Suitable conditions for stresses were also discussed with the labs that were experts for the different pathogens (see Supplementary Table 1), which for most cases provided accurate information based on growth curves and viable counts or from work by others. Due to the vast amount of information, this was not presented in detail in the text. However, in retrospective we can see, that in a few cases it seems that the stresses employed likely had a toxic effect for some conditions/strains, which were excluded (marked with a cross in Fig. 1b). We found that we had to accept some missing information for this comprehensive dataset, which to be of this magnitude, still can be considered as a uniform dataset. Further, since most stress inductions were performed for short periods of time (10-30 min dependent on stress, see Table 1) and are in many cases expected to result in halted replication, growth curves were not considered as an appropriate approach.

For the experiments shown in Fig 3 the authors analyze their data based on the classic distinction by Gram staining. However, this classification might not be the best in reflecting their evolutionary grouping especially for the Gram-negative bacteria. In Fig 1 the authors already group the investigated bacteria by phylogeny. Wouldn't it be better to rather compare their transcriptional response inside, and between the phyla as established in Fig 1? Or would following such a more evolution-based classification with such a relatively low number of investigated organisms make no difference to the presented analysis and results classified by Gram positive/negative?

The assumption of possible variations in overlaps between stress responses of specific bacteria of different phylogenetical orders is accurate. However, the uneven representation of phylogenetical orders regarding number of strains (9 Bacilli, 3 Epsilonproteobacteria, 4 Betaproteobacteria, 14 Gammaproteobacteria, 1 Spirochaetales, and 1 Corynebacteriales strains) makes it difficult to evaluate the suggested phylogeny-based comparisons in a reliable manner. For this, stress responses from more bacterial species would be required.

We did however try to make this kind of analyses to reveal overlaps of stress responses in more specific bacterial groups by re-calculating PTDEX scores for gene groups of G (-) strains in different phylogenetical orders (Epsilonproteobacteria represented by 3 strains, Betaproteobacteria by 4, and Gammaproteobacteria by 14). However, the uneven representation of strains regarding number of species in the different of phylogenetical orders where Gammaproteobacteria members dominated (66%), made it difficult to identify reliable phylogeny-related overlaps of stress responses. G (+) bacteria could not be analyzed in this way, since all G (+) included belong to Bacilli. The influence of this biasness on possible comparisons is now described in the text (line 247-250, page 9).

-The described results about the non-coding sRNA appear a little confusing to me. First the authors note "a substantial proportion of RNAs were transcribed from non-coding DNA regions" and they go on that this is even an underestimation. But how abundant are they really?

The abundance of transcripts from non-coding DNA regions are shown in Supplementary Table 2, where number of total reads, percent mapped reads and % reads mapping to tRNA/rRNA, CDS and intergenic (showing the percentage of reads mapped to ncRNA) are indicated for all samples of all strains in all conditions. The row titles as 'intergenic' is now changed to 'non-CDS'.

And more importantly there are all kind of RNA species and many reasons why they are encoded in these regions. The presence of the more well-known and conserved different types of sRNA's such as csrB interacting and interfering with regulatory proteins such as CsrA as observed in the manuscript is no surprise. And the tmRNA, prominent in Fig 5, is a specific small RNA already known to be important for stress response, since they are crucial for ribosomal quality control allowing to rescue stalled ribosomes and subsequent degradation of the not fully translated proteins in all bacteria. Another interesting sRNA interacting with a protein is the 6SRNA. Is this one detected in the data sets?

We appreciate this comment that inspired us to perform new analyses to detecting 6S RNA transcripts in all samples in the dataset. The resulting data showing proportion of 6S RNA among the total reads mapped the particular genome and also proportion of 6S RNA reads of reads mapping to non-coding regions are now described (line 354-366, page 12-13) in the body text and also presented in new Supplementary Fig. 8.

As the authors mention themselves, there are many possible roles that these various different types small-regulatory RNA's can play. They can act on many targets in trans or as antisense RNA's in cis. For some microorganisms these asRNA species were also reported to be much more abundant than estimated before. Sometimes sRNA's can also encode for small proteins, as the authors also suggest/observe. A question would be, whether these small RNA's can be classified and analyzed comparable to the genes encoding proteins in this study?

Maybe a more in depths analysis of cis and trans acting small RNA, if possible with this approach, could then be a performed in a separate story.

Due to size selection during library preparation as described in the body text (line 325-328, page 11), we cannot confidently assume that our data collection harbor information of all sRNAs expressed under control and stress conditions for any bacterial species. Therefore, unfortunately, the suggested comparison would not be appropriate for this particular dataset. The suggestion is although highly relevant and would likely be of high interest, but due to the nature of the data, a separate study, with appropriate library preparation for sRNAs, would be required.

However, as all the libraries are prepared in a similar way allowing strand-specific read alignments, it is possible for researchers to subtract cis and trans acting sRNAs of the size range allowed by the library preparation to explore these further, as the raw data will be publicly available.

Reviewer #2 (Remarks to the Author):

In an effort to provide a complete and consistent dataset of RNA differentially expressed during stress in pathogenic bacteria, the authors performed transcriptomic analysis of 32 human bacterial pathogens in 11 host related stress conditions using the RNAtag-Seq technique (Shishkin et al, Nat Methods. 2015 Apr; 12(4): 323–325). Appreciably, the bacterial cultures were performed in several laboratories with expertise for each pathogen. They produced an extensive amount of data (1122 transcriptomes in total) which will be made publicly available on the GEO database. They compared gene expression in the different stress conditions and validated their data doing comparison with expression profiles of genes previously known to be regulated under those stresses. In particular, they developed a probability score (PTDEX score) to compare the stress responses between bacteria. To validate the fact that the genes regulated in the stress conditions are relevant during infection, they compared their data with published in vivo datasets from *S. aureus* and *K. pneumoniae*. Finally, they identified genes with altered expression in several stress conditions, which they named "universal stress responders", and proposed that those genes could serve as antimicrobial targets. All the differential expression data in a publicly available database "PATHOgenex", which is easy to understand and use. This database will be helpful for all scientist working on pathogenic bacteria, allowing to check differential gene expression in many stress conditions.

In my opinion the study is clear, well written and the conclusions are supported by the data, however I have some comments regarding the manuscript and suggestions for the database. In general, I would appreciate to know more about the elaboration of the PTDEX score and to have more comments and references about previously published work. Moreover, the paragraph focusing on non-coding RNA could be clarified.

Please note that changes made according to reviewers' comments and suggestions in the revised version of the manuscript are indicated in, yellow-marked, in supplementary marked-up manuscript in supplementary material.

Comments:

*We appreciate this reviewers' in-depth scrutinization of the work. We acknowledge the effort given for simplifying the presentations of complex dataset and findings in this study and also for providing additional information on the existing database for *S. aureus* ncRNAs. The suggestions for making the principles of PTDEX scores more understandable was very much appreciated and we are happy that we could address.*

- Line 78: More comments about Suppl Fig 2a would be welcome in the text e.g. How were these genes selected among known stress regulated genes? Could the authors add references validating the function of those genes in stress response? Was it expected that given genes could be up or down regulated depending on the bacteria? In the legend please define what is the meaning of the crossed squares.

*We have now included new text describing the genes selected as indicative of the stress responses (line 115-133, pages 4-5), now shown in new Supplementary Fig. 3 (previously Supplementary Fig. 2a). Their differential expression profiles are discussed and references showing their relevance for being indicative of the respective stress responses are given, similar is for genes that in some species show opposite regulation. The meaning of the crossed squares, is to show "no differential regulation", which was the case for some of genes in certain species. This is now clarified in the figure legend. Since regulation of *fnr* was quite different in different species and therefore required extensive discussions to not cause confusion, it was removed for fitting word count and reference number limits of the manuscript.*

- Line 83: The authors found that response to bile is significantly higher in G+ bacteria than in G-, is it an overall observation? Or is the variation in only few G+ strains responsible for this difference?

This is a relevant comment, and the observations regarding the response to bile by G+ and G- bacteria are now discussed in the text (138-144, page 5).

- Line 107 and Fig 2b: The authors should elaborate more on the equation – in the text or M&M – explaining precisely how it was derive, if other scores where tested and why this one was chosen. In addition, after explaining the PTDEX score the authors should give detailed information about the meaning of the score.

The thinking behind the equation used for PTDEX calculation is now thoroughly described in the text, separated in the three steps that led to the final equation (line 165-183, page 6).

For instance, what do the lowest and highest values of the score correspond to?
This is now described in text (line 183-186, page 6-7).

They could also give several examples describing different range of scores (or maybe describing how one can obtain the same score for different reason such as maybe important DE with few genes vs moderate DE with many genes). It is written line 153 that when the PTDEX is greater or equal to 0.25 at least 50% of the genes in the group are differently expressed. This information could already be mentioned there to give the reader a more precise idea about the meaning of the score. In addition, the PTDEX score corresponding to the DE expression shown in Suppl fig 2 could be presented.

We provide examples of PTDEX scores together with discussion on different scenarios about gene group conservation and regulation. PTDEX scores of well conserved gene groups vs less conserved gene groups are also discussed (line 189-197, page 7). Moreover, information behind determination of "high PTDEX score" ($\Rightarrow 0.25$) is now given in association to the description about the work with the score, this is also illustrated in Supplementary Fig. 4a, showing examples of gene groups of different sizes and their regulations. Further, a heatmap showing PTDEX scores of the genes selected as indicative of the different stresses (previous supplementary Figure 2a, now Supplementary Fig. 3) are now shown in new Supplementary Fig. 4b.

- Figure 2b and Line 390: In Fig 2b, one parameter of the equation is $n_{\text{species}}(\text{on})$ while in the M&M it is $n_{\text{strain}}(\text{on})$.

We are very thankful for this comment. This was a mistake by us and has now been corrected by ex-changing the wrongly written "strain(on)" to the correct "species(on)" in M&M (line 566, page 19).

- Line 113 and Fig 2d: The authors mention "the expected genes groups that are found in both KO and PGFam groupings", what are those genes? Could the authors describe their function add relevant references?

Genes groups represented in top 20 of both KO and PGFam groupings that are expected to be differentially regulated in low iron and under oxidative stress are now discussed in the text with associated references describing their function and regulation (line 204-207, page 7).

- Fig 3b and 3c: The titles of the heat maps should be revised, both have now the same title while one is comparing G+ and G- responses (for each stress), and the other is comparing the stress responses (in G+ and G-).

We are thankful for this comment and we have changed accordingly. The title of Figure 3b is now 'Similarities between stress responses' and titles in Figure 3c are now 'Similarities between stress responses in Gram-negatives' and 'Similarities between stress responses in Gram-positives'.

- Line 144: Could the authors comment on the overlapping pathways and processes indicated by the KEGG pathways? Giving hints about the relation between the pathways and some examples.
Examples of conserved overlapping pathways such as ribosome biogenesis and amino acid biosynthesis in both Gram-positives and -negatives under certain stress conditions that has significant impact on growth rate are now discussed (line 239-243, page 8). Also, regulation of genes involved in Purine pyrimidine metabolism are shown as overlapping for Li-Oxs-Ns conditions (line 243-247, page 8-9).

Suppl Fig 3: It would be useful to also define USR in Fig. legend.

'USR' is now defined in the legend for Supplementary Fig 6 (previous Supplementary Fig. 3).

Paragraph starting at Line 147:

A total of 421 USR with a PTDEX score greater or equal to 0.25 were retrieved, but how many different genes were considered in total for this analysis?

The number of total individual genes (12 188) for 421 USRs is now indicated (line 256, page 9).

-It seems that only 609 PGFams groups were selected for this analysis, therefore it is expected to find genes involved in basic biological processes and this should be discussed in the text. Maybe it would also be of interest to find the top stress responders specific for each strain, those could pinpoint to strain specific putative antimicrobial targets.

This is a very good idea, especially regarding the possibility to find targets for more narrow antimicrobials. We have therefore retrieved species-specific USRs for all species, now described in the text (line 267-274, page 9-10) and listed in new Supplementary Table 8.

We first identified strain-specific USRs. However, strain-specificity gave very low number of genes for E. coli, H. pylori, and S. aureus as the database contains 3, 2, and 2 strains for those species, respectively, and only one strain for the other species. The resulting gene lists would thereby be affected by this unequal strain distribution and therefore, we listed species-specific instead of strain-specific USRs, which is in parallel with the reviewer's suggestion.

Line 180: Could the authors compare the proportion of non-CDS mapped reads with previously published data? The fact that a large fraction of transcripts originates from non-CDS has been published previously, as many publications report massive antisense transcription in bacterial pathogens.

This is a relevant comment and we have now changed in the text. As suggested we give reference for a previous report (333-334, page 12). As we have not included cis-antisense RNA encoded on CDS opposing strands (now indicated in line 324-325, page 11) such comparisons are not applicable in this study.

- Line 191: The sentence is not clear. The authors state that the level of tmRNA is not high for all strains, for example K. pneumoniae: however, according to Fig. 4b, in this strain the percentage of tmRNA compared to total is one of the highest one (including in the control condition). The end of this paragraph is unclear, as well as the title of Suppl. Fig. 4.

We agree, and the respective sentences are now written in a clearer way, indicating that the percentage of tmRNA was not high for stationary phase condition in Klebsiella (line 342-345, page 12). The two additional paralogs that were mentioned in Supplementary 7a (previous 4) is now mentioned in the body text (line 346-347, page 12).

Did the authors check that the intergenic region whose expression is increased during infection corresponded to an already published sRNA? (S. aureus database: <http://srd.genouest.org/>).

We thank this reviewer for this very useful suggestion that helped to identify the long ncRNA, whose expression increased during infection from the S. aureus database. We found that the long ncRNA is SRS42, and this information is now given in the text (line 350-351, page 12).

- Suppl Table 2: Could the authors comment on the fact that for some samples the percentage of mapped reads dropped below 50%. Do they think that those samples have a quality comparable to that of the other samples?

We appreciate the reviewer's very thorough review that could notice those samples. Indeed, 18 libraries (from 6 different strains and 4 different conditions) of the 1 122 libraries have a percentage of mapped reads below 50%. Since this was seen in all replicates of one sample, it was 6 samples out of 374 with low percent. We could however not observe any major differences in read quality in comparison to other libraries regarding coverage over transcript length and there was no bias towards any of the RNAtag-seq barcodes used during library preparation as same barcodes were used for the replicates of the same stress conditions in all species (see M&M section line 476-478 and Supplementary Table 2). We have also revisited our RNA quality assessment done on agarose gels, but could not observe any difference in RNA quality. Nevertheless, we find the number of total mapped reads in those samples high enough (far exceeding 3M reads, except Salmonella Sp samples with average of 2.7M reads) to cover the whole transcriptome, which should not affect the overall analyses and conclusions.

As the authors pinpointed, many regulated transcripts correspond to non-coding RNAs. It would be valuable to add data for the published small RNAs in the PATHOgenex database.

This was also suggested by Reviewer #1's and we understand that it is attractive to include also non-CDS RNAs. However, due to size selection during library preparation as described in the body text (line 325-328, page 11), we cannot confidently assume that our data collection harbor information of all sRNAs expressed under control and stress conditions for any bacterial species. Therefore, unfortunately, the suggested comparison would not be appropriate for this particular dataset. The suggestion is although highly relevant and would likely be of high interest, but due to the nature of the data, a separate study, with appropriate library preparation method, would be required.

However, as all the libraries are prepared in a similar way allowing strand-specific read alignments, it is possible for researchers to subtract cis and trans acting sRNAs of the size range allowed by the library preparation to explore these further, as the raw data will be publicly available.

In the PATHOgenex database one condition is "Mig" while it is "Hyp" in the manuscript.

This was a mistake and has now been corrected.

Did the authors considered the possibility to also give access to read coverage in a genome browser?

If not, they should give example of read coverage for some genes per bacteria so that one can visualize sequencing data and assess the quality.

We have now provided information of length coverage for gene groups of different sizes for each strain, shown in new Supplementary Fig. 2. For each strain, we show 4 CDSs groupings of different sizes, plotted as normalized coverage over the normalized length of the transcripts. To ensure that we do not miss single samples of bad quality, we used the library with the lowest sequencing reads (as the poorest in coverage) for each species. Homogenous coverage along the transcript length showed the quality of our read mappings. This is now described in the result section, line 107-110, page 4.

In addition, it would be relevant to discuss already existing databases (transcriptomic data of human pathogens available in RNA sequencing genome browsers): e.g. H. pylori (<http://hpylori-tss.imib-zinf.net/>), Salmonella (http://bioinf.gen.tcd.ie/cgi-bin/jay/salcom_v2.pl?_HL, <https://salmonella.wadsworth.org/>) etc.

The suggested databases are now discussed in the introduction, line 71-82, page 3.

REVIEWER COMMENTS

Reviewer #1 (Remarks to the Author):

All my comments and questions were addressed and implemented in the revised manuscript "RNA Atlas of Human Bacterial Pathogens Uncovers Stress Dynamics Linked to Infection" by Avican et al.

I have just some more specific comments, questions or suggestions to the new text:

-p5 l104 "bias" instead of "biasness"

-p6 l169-170 " This calculation gives a value regardless of large (well-conserved) or small (less conserved) gene groups, hence not informative regarding how well this regulation is conserved among the species in the database. "

This sentence is not that clear to me. What is meant by "informative"? maybe "contains no information", "hence it is not informative" or "gives no indication"?

-p9 l263-6 "In accordance, nrdD and nrdG genes encoding Class III ribonucleotide reductase, critical for regulating the deoxyribonucleotides pool required for DNA synthesis and repair, has high PTDEX score in all conditions has indeed been suggested as a target for compounds to inhibit cell growth"

"critical" and "deoxyribonucleotide pool" Maybe: ... displays a high PTDEX score under all conditions and this enzyme class has indeed been suggested as a target for compounds to inhibit cell growth.

-p11 l318 "in vitro" instead of "in vivo"?

-p11 l324-5 Maybe: It is noteworthy that this analysis did not include transcription of cis-antisense RNA encoded on CDS of the opposing strands.

What is meant by "opposing strand"? Maybe "opposite"?

-p11 l326-7 ..."since most sRNAs were excluded during sample preparation due to the >100 nt cut-off for library preparation."

Maybe: ..."since probably many sRNAs..."

-p13 l 377-8 " Other applications are retrieval of most regulated gene groups under certain stress condition(s) in a wide range of bacterial pathogens "

What does "are retrieval" in/and this sentence mean?

Reviewer #2 (Remarks to the Author):

The revised version of the manuscript has addressed the comments raised during the reviewing process. This includes a full description of the PTDEX score elaboration as well as the addition of references and a clearer description of the results.

Regarding species-specific stress responders, I would not qualify those as "universal" stress responders.

- line 125: "paralogues", "upregulation of ahpC"

- line 129: "was upregulated in the majority"
- line 264: "critical"

Anaïs Le Rhun

Reviewer #3 (Remarks to the Author):

Per the editors request, my review focuses on the PTDEX score and differential expression analysis.

Based on the reviewers comments and authors responses to them, it seems like significant effort was made to better explain the logic and algorithm underlying the PTDEX score. However, highlighted by the unaddressed requests from a reviewer that authors explain "if other scores were tested and why this one was chosen", it is still difficult to understand the need for developing this scoring method de novo or how this method is an advance on existing approaches. Specifically, there are several widely-used approaches for determining the statistical significance of observing a proportion of genes in a functional group differentially expressed, including the hypergeometric test and Kolmogorov Smirnov (K-S) test, that take into account some of the same parameters as the PTDEX score, such as what proportion of the genes are differentially regulated under a particular stress condition and what the total number of genes are regulated, but likely are as good or better at assigning probabilities to these observations than the more simplistic algorithm used here. These could be computed for each PGfam or KO per strain per conditions and integrated into other probabilistic models that take into consideration the conservation of genes within and across different phylogenetic groups to compare stress responsive pathways among different strains, species, or other taxonomic groups. In addition, the authors state that "the PTDEX scores of gene groups that are highly conserved among the 32 strains but not regulated in many would have similar PTDEX score with gene groups that are less conserved but regulated in a high proportion of strains harboring the gene". This raises concern that by combining conservation and functional group enrichment into a single score, some functional groups with relatively low (and potentially not statistically significant) enrichment in differentially expressed genes would get a boost from being well conserved that would cause them to be associated with a response to a stress over a different set of genes that are poorly conserved but are highly responsive to that condition. Since the PTDEX score is the foundation of most of the analysis in the study, it is imperative that it is built in a way that better leverages proven algorithms or if it doesn't, that the need for and impact of employing these more ad hoc mathematical approaches be better explained and justified.

It is also puzzling why the authors did not use more established approaches for their differential expression analysis such as DESeq and edgeR, which employ more robust algorithms for analyzing variance in RNA-Seq data and are publicly available. Since all downstream analysis is built on these pairwise gene expression comparisons, conducting these in the most rigorous way is key to ensuring the validity of the authors results and conclusions. Also, while a p-value cutoff makes sense to ensure only changes in abundance that are statistically significant in the context of the variance in the dataset are included, imposing a minimum cutoff of fold change may not be justified here. For example, assume there are 32 genes in a functional group all of which have increased in abundance in a given stress condition compared to the control by 1.4-1.47 fold with p-values all below 0.05 while in another condition 7 of those genes are up 1.7-1.9 fold with p-values below 0.05 while the others went down. The observation that 32 functionally related genes went up in a statistically significant way even at a fairly small magnitude would seem to reflect a coordinated regulation of a pathway even more so than seeing only 7 of those genes go up by a slightly higher magnitude. Yet applying the fold change cutoff would lead to the conclusion that in scenario 1 there was no enrichment of genes in this functional groups that are regulated under the stress condition while in scenario 2 there was. The fold change cutoff is useful in removing single genes whose low magnitude of variance across conditions is unlikely to reflect an actual

physiological response. However, as shown in the example above, when looking at the dynamics of groups of genes, this cutoff may obfuscate relatively small but highly coordinated regulatory responses in pathways that are central to responses and adaptations. This cutoff is fairly low and removing it may not change much in the results but the authors should better explain why they chose to use it, or better yet, present some data suggesting it is necessary to avoid spurious results.

Finally, while the PATHOgenex website is a very useful and accessible tool for interacting with the data at a small scale and the raw data is available through GEO, it is not clear how to access other data generated in this study, such as tables of counts/gene or the results of differential expression analysis for each strain:condition. Access to these will be extremely helpful in leveraging the extensive amount of data generated in this study for other systematic analyses.

REVIEWER COMMENTS

Response to all reviewers:

*Due to a data versioning issue, part of the analysis was ran on an outdated set of results of **general PTDEX score for PGFam groups** presented in the previous version of the manuscript. This has now been corrected in the new version of the manuscript and values in the website have been updated. The scale of general PTDEX scores for the heatmaps in Fig 2dc, Supplementary Fig 4b and 9b have been corrected accordingly. This did not lead to a change in the ranking of gene groups, only that the scores became lower. This also led to that the number of USRs become fewer due to the general PTDEX score >0.25 threshold. This is now corrected in Supplementary Fig 6 where USRs with lower general PTDEX scores have been excluded from Fig. 4a. Associated corrections in the main text are marked with green (lines 192-194, line 202, lines 215-216, line 280, line 284, line 287, and line 311).*

Reviewer #1 (Remarks to the Author):

We are happy to hear that we could address all comments and questions of the reviewer and thankful for the effort. The minor changes are now addressed in this revised version of the manuscript.

All my comments and questions were addressed and implemented in the revised manuscript "RNA Atlas of Human Bacterial Pathogens Uncovers Stress Dynamics Linked to Infection" by Avican et al.

I have just some more specific comments, questions or suggestions to the new text:

-p5 l104 "bias" instead of "biasness"

This is fixed now.

-p6 l169-170 " This calculation gives a value regardless of large (well-conserved) or small (less conserved) gene groups, hence not informative regarding how well this regulation is conserved among the species in the database. "

This sentence is not that clear to me. What is meant by "informative"? maybe "contains no information", "hence it is not informative" or "gives no indication"?

This is fixed now.

-p9 l263-6 "In accordance, nrdD and nrdG genes encoding Class III ribonucleotide reductase, critical for regulating the deoxyribonucleotides pool required for DNA synthesis and repair, has high PTDEX score in all conditions has indeed been suggested as a target for compounds to inhibit cell growth"

"critical" and "deoxyribonucleotide pool" Maybe: ... displays a high PTDEX score under all conditions and this enzyme class has indeed been suggested as a target for compounds to inhibit cell growth.

This is fixed now.

-p11 l318 "in vitro" instead of "in vivo"?

We understand that the associated sentence was not clear. Now the text is 'The nitrosative stress related genes differentially regulated in infected lungs involve...'

-p11 l324-5 Maybe: It is noteworthy that this analysis did not include transcription of cis-antisense RNA encoded on CDS of the opposing strands.

What is meant by "opposing strand"? Maybe "opposite"?

This is fixed now.

-p11 | 326-7 ... "since most sRNAs were excluded during sample preparation due to the >100 nt cut-off for library preparation."

Maybe: ... "since probably many sRNAs..."

This is fixed now.

-p13 | 377-8 " Other applications are retrieval of most regulated gene groups under certain stress condition(s) in a wide range of bacterial pathogens "

What does "are retrieval" in/and this sentence mean?

We are sorry for the typo. It is corrected as 'retrieval' now.

Reviewer #2 (Remarks to the Author):

We are happy to hear that we could address all comments of reviewer and thankful for the effort. The minor changes are now addressed in this revised version of the manuscript.

The revised version of the manuscript has addressed the comments raised during the reviewing process. This includes a full description of the PTDEX score elaboration as well as the addition of references and a clearer description of the results.

Regarding species-specific stress responders, I would not qualify those as "universal" stress responders.

We used the term 'universal' to indicate responsiveness of gene groups to multiple stress conditions. While species-specific USRs are specific to species, they are responsive to multiple stressors as USRs. Therefore, we think that term 'universal' is good a fit for species-specific USRs, too. We would also like to keep it similar for the ease of understanding the difference and similarity in comparison to USRs.

- line 125: "paralogues", "upregulation of ahpC"

- line 129: "was upregulated in the majority"

- line 264: "critical"

These are fixed now.

Reviewer #3 (Remarks to the Author):

We appreciate the reviewer's effort on evaluation of the differential expression analysis and PTDEX score algorithm. The comments and suggestions on PTDEX algorithm, especially the suggestion on comparing it to other known statistical tests had been very fruitful to see that PTDEX score algorithm performs better. Therefore, we believe that the comments and suggestions overall had improved our manuscript.

Reviewer #3 (Remarks to the Author):

Per the editors request, my review focuses on the PTDEX score and differential expression analysis.

Based on the reviewer's comments and authors responses to them, it seems like significant effort was made to better explain the logic and algorithm underlying the PTDEX score. However, highlighted by the unaddressed requests from a reviewer that authors explain "if other scores were tested and why this one was chosen", it is still difficult to understand the need for developing this scoring method de novo or how this method is an advance on existing approaches.

We have now indicated in the text that we also used hypergeometric test. We have better explained why the PTDEX method is an advance on existing approaches in our responses to reviewer's comments below.

Specifically, there are several widely-used approaches for determining the statistical significance of observing a proportion of genes in a functional group differentially expressed, including the hypergeometric test and Kolmogorov Smirnov (K-S) test, that take into account some of the same parameters as the PTDEX score, such as what proportion of the genes are differentially regulated under a particular stress condition and what the total number of genes are regulated, but likely are as good or better at assigning probabilities to these observations than the more simplistic algorithm used here. These could be computed for each PGfam or KO per strain per conditions and integrated into other probabilistic models that take into consideration the conservation of genes within and across different phylogenetic groups to compare stress responsive pathways among different strains, species, or other taxonomic groups.

The equation we used to generate the PTDEX score considers the differential regulation of same genes from different species in the PATHOgenex atlas. We aimed to give a score to each gene group for a particular condition that would provide a hint for how same genes from species not present in PATHOgenex atlas can be expected to be regulated under this condition.

Indeed, PTDEX score algorithm is not more simplistic than a Hypergeometric distribution test and/or a Kolmogorov Smirnov test as it considers six different features, including number of genes in each gene group, proportion of regulated genes in each group, number of species harboring at least one gene, number of species having at least one regulated gene, number of strains harboring at least one gene, and number of strains in the database. These features are used to evaluate conservation of the response to each stress condition in all the strains covered by the database, whereas the tests mentioned by the reviewer cannot account for the number of impacted species. Additionally, our score weighs the degree of gene conservation among the tested strains and minimizes the boosting effect of well-conserved genes, and could do better in ranking of particular stress associated gene groups. Nevertheless, we found the reviewer's suggestion interesting and employed the hypergeometric test and Kolmogorov Smirnov (K-S) test on the PATHOgenex dataset for the same purpose of PTDEX score algorithm.

We employed the suggested tests with the proportion of differentially regulated genes and conservation. However, we did not use them in downstream probabilistic models considering phylogenetic information, as the dataset has uneven distribution of strains in different phylogenetic orders. This indeed did not allow us to do comparisons of stress responses between different phylogenetic orders as explained in lines 271-274.

The K-S test was found to rank mostly small gene groups as groups with highest probability to be regulated under a particular stress condition. In many cases, none of the genes in a top ranked gene group was differentially regulated under the associated stress condition. We applied the K-S test on the measured fold change values of gene differential expression and used a normal

distribution as reference distribution. This made the test very sensitive to small groups of 2-5 genes (hence the high scores for those), and made it very affected by the large proportion of non-differentially expressed genes in the considered groups. This, combined with the inability of the test to account for species-specificities, makes it not applicable for our purpose.

The hypergeometric test was more successful in ranking particular stress associated gene groups compared to that seen for the K-S test. We applied the hypergeometric test on count values obtained using the same filtering criteria as for our PTDEX computation. The hypergeometric test was however not as good as PTDEX in ranking particular stress associated gene groups and provided a very low number of statistically significant gene groups (3-8) per condition. The hypergeometric test showed a bias that gives higher probability for more conserved gene groups. For example, the top 20 gene groups ranked by hypergeometric test (without p-value cutoff, since statistically significant gene groups are lower than 20) under low iron, oxidative stress, and nitrosative stress (conditions where the response is expected to be relatively specific to stressors) had mean values of gene groups' size of 19.5, 15.3, and 12.6 respectively. On the other hand, these values were smaller (15.8, 14, and 9.5, respectively) for gene groups ranked by the PTDEX scores. This indicates that PTDEX score is more successful in ranking particular stress associated gene groups than hypergeometric test even for less conserved gene groups. The reasons behind this is explained in detail in the response to reviewer's comment below concerning the boosting effect of conserved gene groups.

In addition, the authors state that "the PTDEX scores of gene groups that are highly conserved among the 32 strains but not regulated in many would have similar PTDEX score with gene groups that are less conserved but regulated in a high proportion of strains harboring the gene". This raises concern that by combining conservation and functional group enrichment into a single score, some functional groups with relatively low (and potentially not statistically significant) enrichment in differentially expressed genes would get a boost from being well conserved that would cause them to be associated with a response to a stress over a different set of genes that are poorly conserved but are highly responsive to that condition.

We agree with the fact that some functional groups (we assume the reviewer meant gene groups here) with relatively low (and potentially not statistically significant) enrichment in differentially expressed genes would get a boost from being well conserved, which would cause them to be associated with a response to a stress over a different set of genes that are poorly conserved but are highly responsive to that condition. This was one of the most challenging features we had to consider when developing the equation and we appreciate that reviewer highlighted it.

Indeed, we have seen this effect to be more pronounced when no filtering was employed on the fold change, while keeping FDR corrected p-value <0.05 and therefore tested different fold change cutoffs such as >1.25, >1.5, and >1.75. It was clear that ranking gene groups with highest PTDEX scores in particular stress conditions (where the response is expected to be relatively specific to stressors such as low iron, oxidative stress and nitrosative stress), resulted in highest numbers of typical stress related gene groups (for a particular stress) among the top 20 using a fold change cutoff of >1.75. This cutoff improved the equation by contributing to higher scores for specific stress related genes and also minimized the boosting effect of well conserved genes.

There are two additional features of the PTDEX score equation that deal with the boosting effect of conservation. First, the nspecies(on) value (instead of nstrains(on) value) in the second part of the equation avoids taking the sum of differentially regulated genes in different strains of a same species. Instead, it uses the value of one for all strains of the same species if regulated. This takes out the effect of the conservation among species of one strain, which in other case would have a non-relevant influence of the final score. Second, in the last part of the equation, it takes the log2 value of number of regulated genes in the gene groups. This was introduced to minimize the effect of high counts for well conserved genes but still keep a degree of difference. This is explained in lines 173-186.

In summary, given what is explained above, we do not agree with the reviewer's comment suggesting that the algorithm used for PTDEX score is more simplistic than a Hypergeometric distribution test and/or Kolmogorov Smirnov test. One advantage of said tests would be to provide indications of statistical significance of the gene groups. That being said, with the significantly high number of performed (124'883 tests for PGFams, 60'280 tests for KOs), we observed that the Bonferroni-corrected significance threshold only leaves us with an average of 3.9 and 4.6 significant groups per stress condition (for PGFams and KOs respectively) when looking at the

hypergeometric test results. Furthermore, it provides very low number of the statistically significant gene groups (p -value $< 2.70032 \times 10e-07$), which we found to be 3-8 per condition.

Finally, the limitations that the hypergeometric and K-S test have in considering the species and strain distributions make them inappropriate for the purpose of our analyses.

It is worth to note that we determined PTDEX score > 0.25 has at least 50% of genes in the gene groups with different conservation level beside many features used in PTDEX scores.

*The conserved genes, with at least 50% being regulated, seems to be associated to many stress responses. As found in USRs, such genes are involved in basic metabolic pathways that are simply affected by the bacterial growth which is known to be restricted under various stress conditions. Therefore, the highly responsive behavior of conserved genes under various stress conditions is not surprising. Nevertheless, the high PTDEX score for conserved genes is still provides a hint on the probability of those genes in other bacterial species, that are not in PATHOgenex atlas, to be regulated under particular conditions. Noteworthy, the PTDEX is not used to identify stress specific genes but rather the probability of gene to be regulated in a particular stress condition. Nonetheless, PTDEX score could be able to identify stress specific genes when the effect of stress producing agent is very specific such as depletion of iron with addition of 2,2'-bipyridyl in the culture. As seen for *sufA* and *tonB* in Fig 2d, these two genes are relatively low conserved but still being top ranked among the genes with high PTDEX score under low iron condition. In accordance, such observation may not be the case for stress condition triggering various stress conditions such as Hypoxia, Nutritional downshift, and Stationary phase conditions.*

Since the PTDEX score is the foundation of most of the analysis in the study, it is imperative that it is built in a way that better leverages proven algorithms or if it doesn't, that the need for and impact of employing these more ad hoc mathematical approaches be better explained and justified.

We hope that we could justify the usage of PTDEX scores in the responses above. We highlighted this in lines 186-191 in the revised version of the manuscript.

It is also puzzling why the authors did not use more established approaches for their differential expression analysis such as DESeq and edgeR, which employ more robust algorithms for analyzing variance in RNA-Seq data and are publicly available.

Differential Expression Analysis in CLC Workbench is detected based on the fit of a Generalized Linear Model (GLM) with a negative binomial distribution, similar to edgeR and DESeq2. The model supports paired designs, as well as multiple comparisons. All pair groups comparison against the control group are done by Wald test similar to DESeq2. Moreover, worth to mention, we have tested both DESeq and edgeR on our data and we could not find a significant difference in the number and composition of differentially expressed genes compared to what was obtained with the CLC system. Therefore, we have used CLC Genomic Workbench to ease the handling, quality control, read mappings, read counting, and visualization of such big dataset.

Since all downstream analysis is built on these pairwise gene expression comparisons, conducting these in the most rigorous way is key to ensuring the validity of the authors results and conclusions.

*We believe that we have ensured the validity of our data in Supplementary Fig. 3 showing that genes known to be regulated by a particular stress condition in different bacterial species were indeed identified as differentially expressed under the particular condition with our analyses. This was also discussed in lines 113-133. This could be extended with many other known regulations of genes such as the genes encoding type three secretion system and its effector proteins under virulence inducing condition in *Yersinia pseudotuberculosis*.*

Also, while a p-value cutoff makes sense to ensure only changes in abundance that are statistically significant in the context of the variance in the dataset are included, imposing a minimum cutoff of fold change may not be justified here. For example, assume there are 32 genes in a functional group all of which have increased in abundance in a given stress condition compared to the control by 1.4-1.47 fold with p-values all below 0.05 while in another condition 7 of those genes are up 1.7-1.9 fold with p-values below 0.05 while the others went down. The observation that 32 functionally related genes went up in a statistically significant way even at a fairly small magnitude would seem to reflect a coordinated regulation of a pathway even more so than seeing only 7 of those genes go up by a slightly higher magnitude. Yet applying the fold change cutoff would lead to the conclusion that in scenario 1 there was no enrichment of genes in this functional groups that

are regulated under the stress condition while in scenario 2 there was. The fold change cutoff is useful in removing single genes whose low magnitude of variance across conditions is unlikely to reflect an actual physiological response. However, as shown in the example above, when looking at the dynamics of groups of genes, this cutoff may obfuscate relatively small but highly coordinated regulatory responses in pathways that are central to responses and adaptations. This cutoff is fairly low and removing it may not change much in the results but the authors should better explain why they chose to use it, or better yet, present some data suggesting it is necessary to avoid spurious results.

We agree with the reviewer that fold change cutoff is useful in removing single genes whose low magnitude of variance across conditions is unlikely to reflect an actual physiological response. Many similar transcriptomics analyses that have been published, commonly use higher fold change cutoffs (Smith et. al, 2016, Kroger et. al, 2013,) then what we have used in this study. As discussed above, it was necessary to employ a fold change cutoff in the PTDEX score calculation in order to retrieve more stress specific gene groups. Even though the reviewer's assumptions on the two scenarios sounds very reasonable, we have seen that not employing fold change cutoff or fold change <1.75 is far from giving meaningful output on PTDEX score as mentioned above. It was clear that no or low cutoffs filtering lead to undeserved high score 's for well conserved genes and masks other stress-associated but less conserved genes.

The data from analyses using different levels cutoffs for differential expression and PTDEX calculations lead to that we employed a fold change >1.5 for differential expression analyses to do not miss any differentially expressed genes in each stress condition for inter-species/stress comparisons and >1.75 for definition of ngenes(on) in PTDEX score calculation due to the reason mentioned above.

We are very thankful to the reviewer for highlighting this point that made us realize that we had missed to include the information that the ngenes(on) definition is genes with FDR-p-value<0.05, fold change >1.75, and maximum group expression >= 20 in the material method section. This information is now included (lines 567-568).

Further, inspired by the arguments regarding possibilities to receive additional information depending on the cut-off used we have now introduced a function in the online PATHOgenex atlas that allow users to set different fold changes cutoffs together with p-value.

Finally, while the PATHOgenex website is a very useful and accessible tool for interacting with the data at a small scale and the raw data is available through GEO, it is not clear how to access other data generated in this study, such as tables of counts/gene or the results of differential expression analysis for each strain:condition. Access to these will be extremely helpful in leveraging the extensive amount of data generated in this study for other systematic analyses.

The TPM values of each gene under each stress condition is also deposited in GEO. These data will be publicly available upon publication of the manuscript. The TPM values and differential expression of each single gene under each stress condition is also available in PATHOgenex RNA atlas (www.pathogenex.org). The raw data with accession numbers of reference genomes are sufficient to generate the read count tables and perform differential expression analysis for each strain and condition. However, as an additional service, we have now included an option in PATHOgenex atlas webpage that the processed metadata read count tables and differential expression analyses will be available upon request.

REVIEWERS' COMMENTS

Reviewer #1 (Remarks to the Author):

All my comments and suggestions were addressed.

Reviewer #2 (Remarks to the Author):

I do not have any further comments/suggestions. The manuscript is suitable for publication.

Anais Le Rhun

Reviewer #3 (Remarks to the Author):

The author's responses and revision address the comments and concerns raised in my previous review.

Jonathan Livny